# The airborne transmission of viruses causes tight transmission bottlenecks

Patrick Sinclair[1], Lei Zhao [2], Clive B. Beggs[3] & Christopher J. R. Illingworth [1] ✉

The transmission bottleneck describes the number of viral particles that initiate an infection in a new host. Previous studies have used genome sequence data to suggest that transmission bottlenecks for influenza and SARS-CoV-2 involve few viral particles, but the general principles of virus transmission are not fully understood. Here we show that, across a broad range of circumstances, tight transmission bottlenecks are a simple consequence of the physical process of airborne viral transmission. We use mathematical modelling to describe the physical process of the emission and inhalation of infectious particles, deriving the result that that the great majority of transmission bottlenecks involve few viral particles. While exceptions to this rule exist, the circumstances needed to create these exceptions are likely very rare. We thus provide a physical explanation for previous inferences of bottleneck size, while predicting that tight transmission bottlenecks prevail more generally in respiratory virus transmission.

The SARS-CoV-2 pandemic sparked a broad range of interest in both the mechanism and the risks of viral transmission. Early in the pandemic, the mechanism of viral transmission was a matter of controversy, with a claim that transmission occurred either via contact or by the short-range spread of emitted droplets omitting the potential for longer-range airborne transmission[1]. Subsequent work highlighted the importance of aerosolised particles in causing long-range airborne transmission[2] while downplaying the importance of contact-driven events[3].

Studies of the risk of transmission examined the relationship between transmission and the environment, with for example higher rates of transmission being found in household compared to workplace environments[4]. Quantitative models were developed, assessing the risk of infection in a different scenarios[5–8], modelling the relationship between risk and exposure time[9], and explaining the role of masks in preventing viral spread[10]. $CO_2$ monitoring was suggested as a means to evaluate the immediate risk of transmission[11].

While risk calculations consider whether a person might be infected, evolutionary biology poses a different question: If a person was infected, how many viruses initiate that infection? This number of viruses, denoted the transmission bottleneck[12], has important consequences for virus evolution: The tighter the bottleneck, and the

fewer particles get through, the less genetic diversity will be transmitted between individuals. The absence of initial diversity can limit the potential for within-host evolution, as variants need to be generated de novo before evolutionary changes can take effect[13].

Studies of genomic data have suggested that for influenza and SARS-CoV-2 infection, the transmission bottleneck generally involves few viral particles[14–17], with potentially a single virus initiating infection. Different genomic approaches have been applied to this question. In animal models, barcoded viruses allow for a straightforward count of the number of viruses initiating infection[18]. Where barcoding is not possible, deep sequencing of a viral population has been used to assess the appearance or non-appearance of minor variants following the bottleneck[19] or to evaluate changes in variant frequencies during the transmission process[20–23]. Genomic studies have some limitations. Collecting genomic data from individuals is time-consuming and expensive, while the results of such studies may reflect only the specific circumstances of the individuals involved. The estimation process itself requires some care: the false identification of variants has the potential to inflate the estimated bottleneck size[24,25]. Errors in identifying who infected who could also potentially distort results.

We here take an alternative approach to estimating respiratory virus transmission bottleneck sizes. Rather than considering only

[1]MRC University of Glasgow Centre for Virus Research, Glasgow, UK. [2]Section for GeoGenetics, Globe Institute, University of Copenhagen, Copenhagen, Denmark. [3]Carnegie School of Sport, Leeds Beckett University, Leeds, UK. ✉e-mail: christopher.illingworth@glasgow.ac.uk

specific circumstances or data, we outline a general solution, exploiting knowledge of the physical processes underlying viral transmission[26,27] to build a physical model of virus transmission. Within this model, we exploit knowledge from an extensive past literature[28]. Coughing, speaking and sneezing have been shown to emit broad and distinct distributions of particle sizes[29–31]. Emitted particles are affected by evaporation, sedimentation and diffusion[32]. Ventilation reduces the mean concentration of particles in the air, while in the absence of immediately finding a new host, viruses in emitted particles begin to decay[33]. Combining insights from this literature, we assess the expected transmission bottleneck for infections that occur under a variety of scenarios. Our results provide a strong indication, independent of genome sequence data, that most cases of respiratory virus transmission will involve a tight population bottleneck.

## Results

### Constant levels of exposure

A simple model, based upon a Wells-Riley model of exposure, suggested that transmission bottlenecks arising from exposure to a respiratory virus are likely to be tight (Fig. 1). We made the simplifying assumptions that all individuals in an environment receive the same level of viral exposure and that viruses cause infection independently of one another. Under these assumptions, bottleneck sizes are small unless a very large proportion of individuals present in an environment are infected. The Skagit choir superspreading event was an extreme

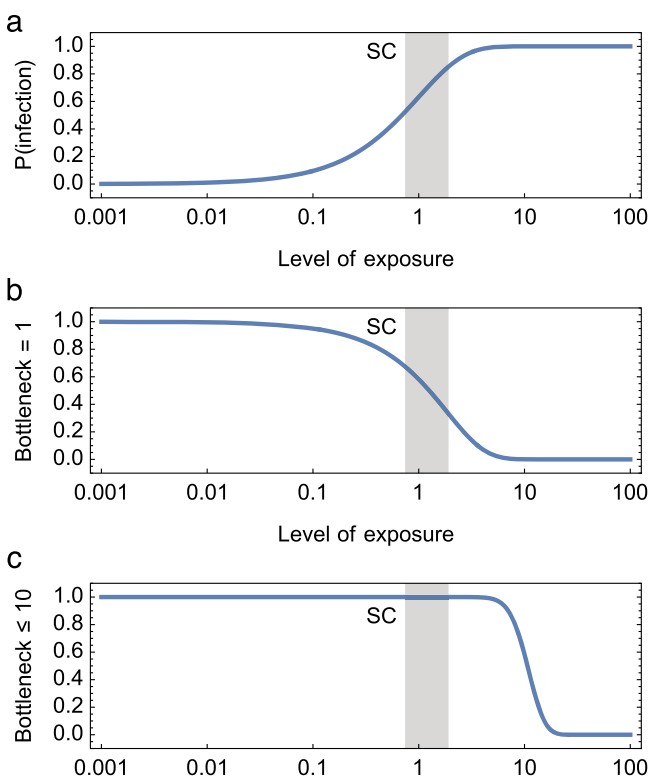

**Fig. 1 | Transmission bottleneck estimates under a Wells-Riley model of exposure.** In this model, the level of exposure describes the rate parameter of a Poisson model, such that a person receiving an exposure of 1 would expect to be infected by one virus. **a** The probability of an individual being infected given the level of exposure. **b** The proportion of cases of infection in which a single virus initiates infection. **c** The proportion of cases of infection in which ten or fewer viruses initiate infection. The vertical grey bar provides an estimate of circumstances at the Skagit Choir superspreading event, characterised by the probability of infection. Even under these circumstances, the model suggests that transmission bottlenecks are likely to be small.

case early in the SARS-CoV-2 pandemic whereby between 32 and 52 of 61 people present at a choir rehearsal were infected[34]. Even in this extreme case, our model predicted that between 33% and 75% of cases of infection were initiated by a single viral particle, with more than 99% of cases being initiated by fewer than 10 viruses.

### Variable levels of exposure

More complex models of exposure produced similar results, suggesting that the transmission bottlenecks produced by respiratory infection are generally tight. A straightforward approach to expanding our initial model is to incorporate overdispersion into the exposure levels; this did not substantially change the results obtained (Supplementary Fig. 1, Supplementary Note 1). To achieve a more realistic estimate of the extent to which exposures vary, we implemented a physical model of virus transmission. Our model describes the emission by an infected individual of virus-containing particles with a distribution of sizes, by default modelling a process of coughing. Emitted particles may contain more than one virus, according to their size. Particles are subject to evaporation and spread through the air by diffusion. They are lost from the air due to ventilation and sedimentation. Viruses within particles are inactivated over time (Fig. 2a).

Levels of physical exposure were calculated based on estimated inhalation rates and then converted into viral exposures (Fig. 2b). Our model describes an effective viral load, defined as the number of viruses per ml of emitted material that initiate infection, having overcome the various barriers, whether physical or immunological, to achieve this. The effective viral load is by nature smaller than the absolute number of viruses contained within an emitted particle: One study has estimated the proportion of emitted SARS-CoV-2 viruses that are viable (measured experimentally via plaque- or focus-formation in cells facilitating infection) as roughly 1 in 3000[35]. Plaque formation is likely a necessary requirement for a virus to cause infection but may not be sufficient: A virus that would form a plaque under laboratory conditions might not be able to cause infection in a host.

Applied to four different environments and run under default parameters, our model suggested that respiratory viral infection arising following coughing is associated with a tight transmission bottleneck. Clear environmental impacts upon exposure were evident, with the highest exposure occurring at close proximity in the poorly ventilated lounge. In the bus, a very broad distribution of exposures was found, with the simulated absorption of particles by the sides of the bus leading to low exposures far from the infected person (Supplementary Fig. 2). Transmission bottlenecks were not universally tight: One of the multiple simulations we generated describing the nightclub environment included a case where 391 viruses initiated infection. However, in all the environments and under our default parameters, more than 98% of transmission events were predicted to involve ten or fewer viruses, with the majority of cases of infection being initiated by a single viral particle (Fig. 3).

The outputs from our model include a parameter, $R_{env}$, describing the expected number of cases of infection occurring in each environment. This parameter is akin to the common epidemiological parameter $R_0$. Where $R_0$ describes the expected total number of infections caused by an infected individual during the entire course of an infection in the absence of population immunity[36], $R_{env}$ describes the expected number of infections caused by an infected individual in a specific environment, given the number of uninfected individuals present, their relative positions, the length of time spent in that environment, and the prevailing environmental conditions. Under our default parameters, these numbers were generally small, ranging from 0.058 in the bus to 0.271 in the nightclub, reflecting the limited time modelled in each scenario. The value of 0.067 in the office environment is a feature of our default model, calculated from the $R_0$ value of the original Wuhan strain SARS-CoV-2 virus (see Supplementary

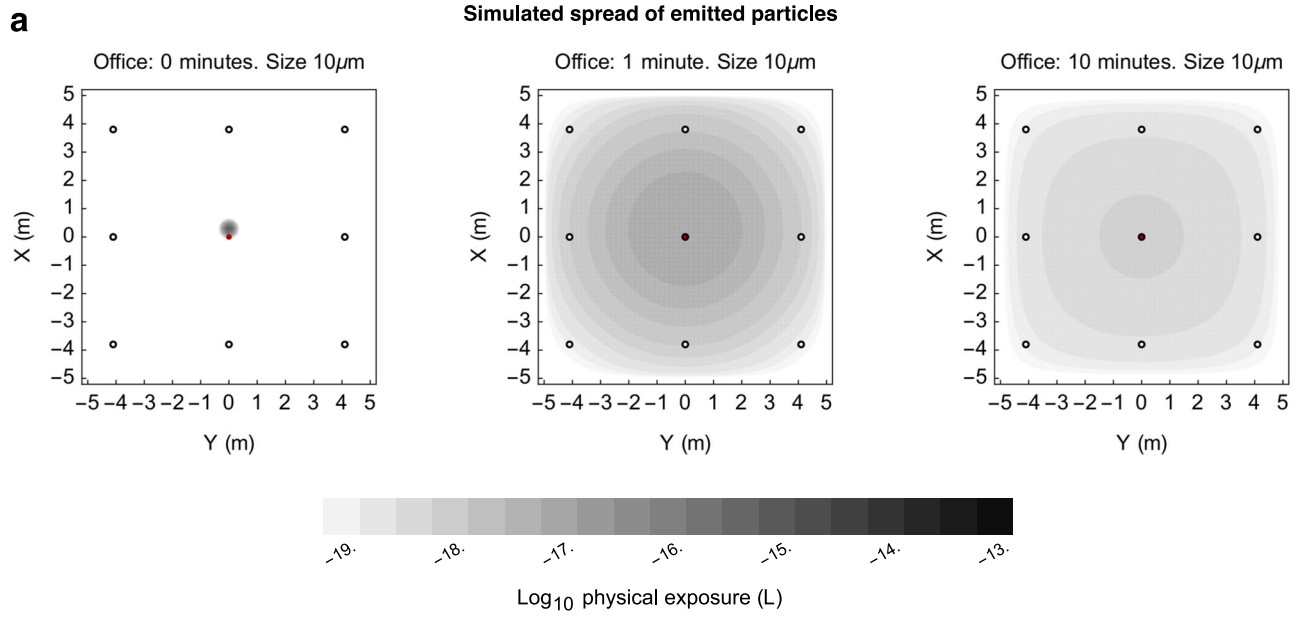

**a**

Simulated spread of emitted particles

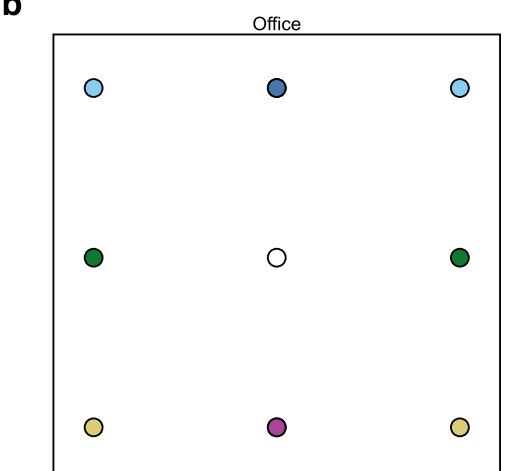

**b**

Office

Sum exposures of individuals across time and particle sizes

| Physical exposures (pL) | | Viral exposures |
|---|---|---|
| 0.943 | | 3.02 x 10⁻² |
| 0.173 | $R_{env}$ = 0.0669 | 5.53 x 10⁻³ |
| 0.572 | Effective viral load: 3.204 x 10⁷ /ml | 1.83 x 10⁻² |
| 0.139 | | 4.44 x 10⁻³ |
| 0.676 | | 2.16 x 10⁻² |

**Fig. 2 | Method for simulating transmission events. a** A computational model described the emission and subsequent dynamics of virus-containing particles following a single cough. We modelled the diffusion of particles of different sizes through space and time, accounting for evaporation, sedimentation, ventilation, and the inactivation of viruses within infectious particles. Our model describes the time- and location-dependent concentration of infectious material within an environment. **b** Our model facilitates the calculation of the cumulative volume of infectious material that we would expect for different individuals in an environment. Specifying an effective viral load, or alternatively the parameter $R_{env}$, which describes the expected number of infections to occur within an environment, generates viral exposures, which describe the expected number of infectious viruses that initiate infection within each person: The outcome of exposure, whether infection or non-infection, is characterised by this viral exposure.

Methods 1). Keeping the level of physical exposure constant, an increase in the effective viral load leads to an increase in $R_{env}$.

**Large bottlenecks at very high effective viral load**
Under our default model, tight transmission bottlenecks were inferred to dominate in all but exceptionally high values of the effective viral load (Fig. 4a, b). Most transmission events involved 10 or fewer viral particles unless the effective viral load was greater than $10^{9.2}$ per ml. This value is greatly in excess of an estimated upper bound for the number of plaque-forming units at peak viral load during SARS-CoV-2 infection[35]. At this concentration, high transmission bottlenecks occur following the inhalation of even a single emitted particle: A particle of radius 10 μm would be expected to contain more than 6 effective viruses.

In most of the environments we simulated, one person coughing was not sufficient for everyone present to inhale a single emitted particle. As such, not everyone in the environment was infected, even at extremely high simulated effective viral loads (Fig. 4c). Alternative models described higher levels of particle emission. For example, a model of continuous, uninterrupted speech, while allowing for asymptomatic transmission, generated higher volumes of particles than did coughing. Increased volume excepted patterns of physical exposure from speaking were similar to those derived from coughing (Supplementary Fig. 3). The increased volume emitted led to an increase in the values of $R_{env}$ in each environment. Compared to the coughing model, a decrease in the proportion of infections initiated by a single viral particle was inferred for the lounge environment, but otherwise, results were very similar (Supplementary Fig. 4). Most

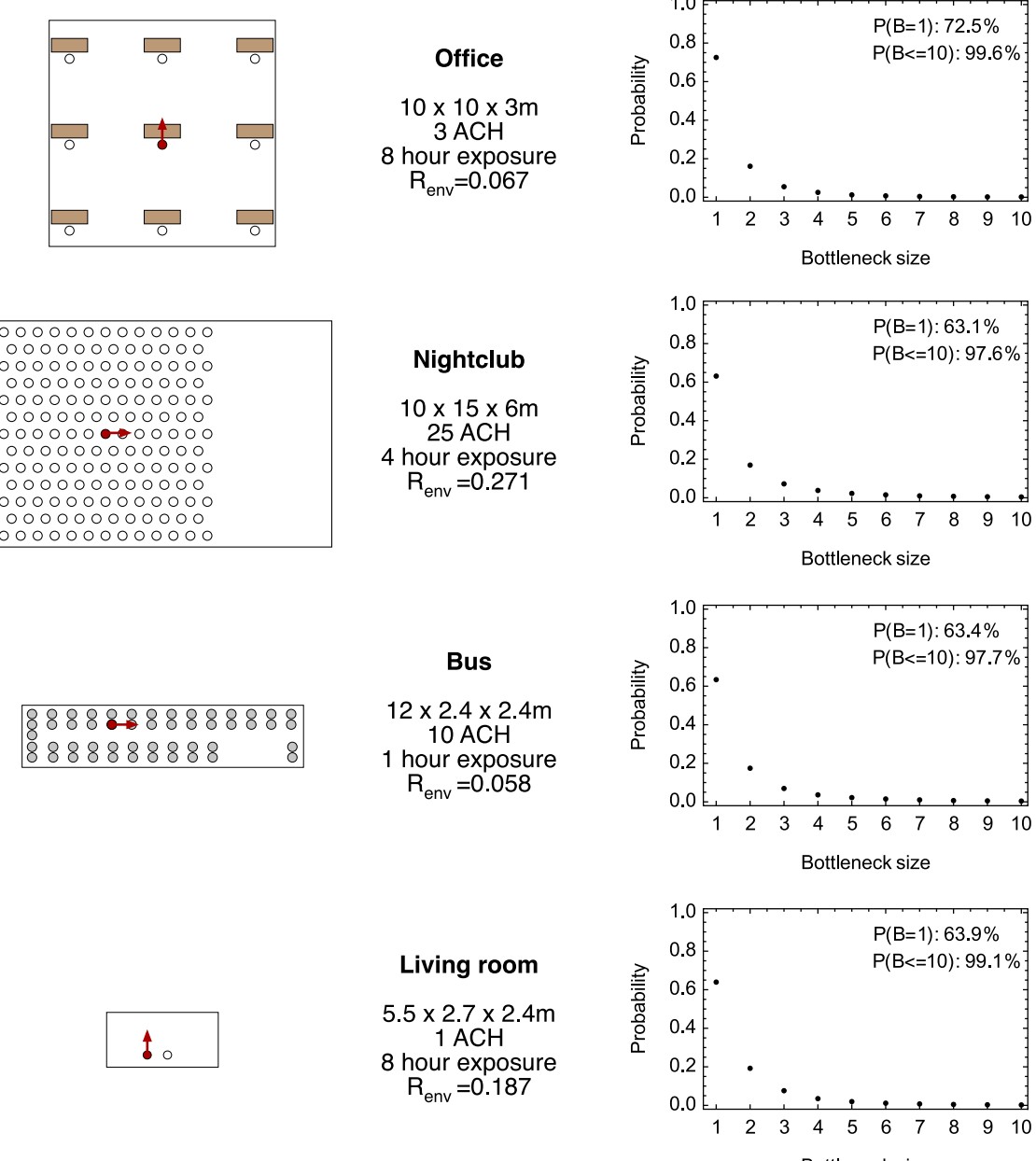

**Fig. 3 | Bottleneck size distributions calculated for different scenarios.** Maps show the layouts of different environments. A red dot indicates the location of an infected person, with the red arrow showing the direction in which emissions occur. A white dot indicates the location of an uninfected person. In our model, individuals were assumed to remain stationary. Furniture did not affect the model and is shown for purely illustrative reasons. Data show the room dimensions. Ventilation levels are described by the number of air changes per hour (ACH). The value $R_{env}$ describes the expected number of people infected in an environment during the modelled time of exposure. Bottleneck size distributions show empirical probabilities calculated from an ensemble of $10^6$ simulations generated for each environment.

transmission events still involved 10 or fewer viral particles unless the effective viral load was greater than $10^8$ per ml, again substantially above published estimates for SARS-CoV-2 infection (Supplementary Fig. 5).

**Large bottlenecks during extreme superspreading**

Modelling identified a second scenario in which larger bottlenecks could prevail. If a highly effective viral load is combined with a very large volume of infectious material is emitted, most infections are initiated by more than 10 viral particles. We generated variants of the coughing model in which the volume of particles emitted was arbitrarily increased. At exceptionally high volumes of emission, the need for individuals present to inhale an emitted particle is no longer a

consideration; nearly everyone present receives some physical exposure. This implies that, above a threshold effective viral load, everyone is likely to be infected, while at some higher threshold effective viral load, everyone is likely to be infected by more than 10 viral particles. The exact values of thresholds were environment-dependent, being affected by the distribution of physical exposures.

With a 1000-fold increase in emission volume, simulation data suggested that most infections in the office environment were likely to be initiated by more than 10 viral particles if the effective viral load was in excess of $10^{7.5}$ per ml (Fig. 5). At this threshold $R_{env}$ was close to the hypothetical maximum value of 8, such that everyone present was highly likely to be infected. For the nightclub, a threshold effective viral load implied a $R_{env}$ value close to 100. Our lounge environment was

a

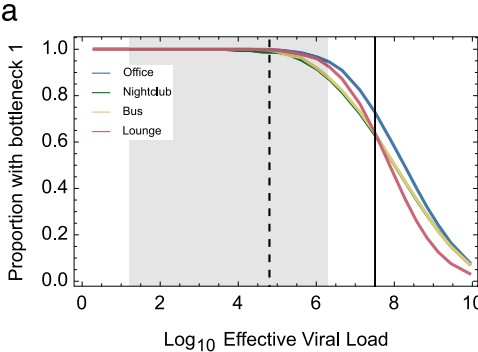

b

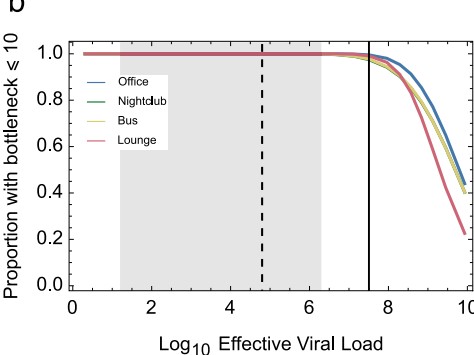

c

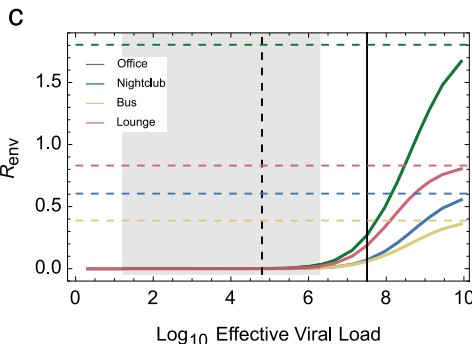

**Fig. 4 | Inferred statistics of transmission bottleneck sizes given changes in the effective viral load.** Statistics were calculated from an ensemble of $10^6$ simulations for each combination of environment and effective viral load. Lines connect points calculated at different viral loads. The dashed vertical black line shows the mean number of plaque-forming units at the peak of SARS-CoV-2 infection, while the grey shaded area shows a 95% confidence interval for this statistic[35]. The solid vertical black line shows the effective viral load as specified in the default parameters of our model. Statistics are shown describing (**a**). The proportion of transmissions with bottleneck size 1. **b** The proportion of transmissions with bottleneck size 10 or less. **c** The expected number of people infected in each environment, $R_{env}$. Horizontal dashed lines in this figure indicate limit values for each environment, as would occur given a theoretically infinite effective viral load.

designed to maximise physical exposure, with a single individual being in prolonged short-range proximity to an infected individual in an environment characterised by poor ventilation. At 1000-fold increased emission volume, most infections in the lounge were likely to be initiated by more than 10 viral particles at an effective viral load of close to $10^6$, still high but within biological plausibility. In the bus the high level of variation in physical exposure levels meant that 1000-fold increased emissions were still not sufficient to infect everyone present.

This high-emission, high-viral load scenario represents exposure to overwhelming numbers of viruses. The identification of super-spreading events early in the SARS-CoV-2 pandemic[37,38] suggests that such a scenario could be biologically plausible. Behaviours such as

singing or shouting would generate higher volumes of emission than our model of speech[39]. However, basic epidemiology suggests that these events are rare: the single-figure values of $R_0$ associated with most respiratory viruses are not compatible with a situation in which infected people transmit the disease to the majority of their contacts.

## Conclusions

Concluding our analysis, we note that individual cases of transmission involving multiple viral particles may arise under unspectacular circumstances: In our default model between 1 and 3% of transmissions involved more than 10 viruses. However, a scenario in which the majority of transmission events involve more than 10 viruses is unlikely, requiring a very high effective viral load, possibly combined with abnormally high levels of particle emission. Our model is parameterised in a way that is consistent with SARS-CoV-2 infection but is not specific to that virus. Where a virus is spread by respiratory transmission, if $R_0$ is not exceptionally high, the physical process of airborne transmission will lead to mostly tight transmission bottlenecks.

Our model may be elaborated in a variety of ways, considering, for example, the movement of people within an environment, emissions via sneezing, changes in ventilation levels, or variable levels of infectivity. None of these changes produced substantial changes in our basic results. Details are given in Supplementary Note 2.

## Discussion

We have here applied two distinct modelling approaches to consider the transmission bottleneck sizes generated by respiratory viral transmission. In a first, highly simplified approach, we showed that, in a case where all exposed individuals receive an equal level of exposure, the Poisson assumption underlying the Wells-Riley model implies that the great majority of transmission events involve a small number of viral particles, even in cases where a high proportion of individuals present are infected. We next considered a more complex, though still approximate method, in which different individuals received different exposures, calculated from a physical model describing particle emission and spread. In this latter model, we identified that the airborne transmission of viruses is dominated by tight transmission bottlenecks in all but two cases. Firstly, if the effective viral load is sufficiently high, the inhalation of a single emitted particle will result infection by several viruses so that transmission bottlenecks will be high irrespective of the level of exposure. Secondly, where the effective viral load and the volume of emitted particles are both very high, most cases of infection will again involve large numbers of viruses; this latter case is associated with a high proportion of individuals present being infected. We believe that each of these two cases represents rare circumstances. In the former case, the effective viral load needed is greatly in excess of a published estimate of the number of plaque-forming units of SARS-CoV-2 at peak infection. In the second case, the very large number of infections generated by transmission is so high as to imply either that these circumstances are very unusual or that the virus has an extremely high value of $R_{env}$ and, therefore, of $R_0$.

The results of our model are consistent with previous studies of transmission bottlenecks that have used viral genome sequence data. For example, a study of influenza virus transmission suggested that between 28 and 31 (73–82%) of a set of 38 transmission events were likely to have been founded by a single virus[15,40]: Under default parameters, our model produced similar results.

Our approach is distinct from previous work in that, not relying upon genomic data describing any particular virus or circumstance, our result is a general one. In this sense, our work is predictive: If there were to be an outbreak of a novel virus spreading by airborne transmission, our model suggests that the transmission of that virus would be characterised by tight transmission bottlenecks.

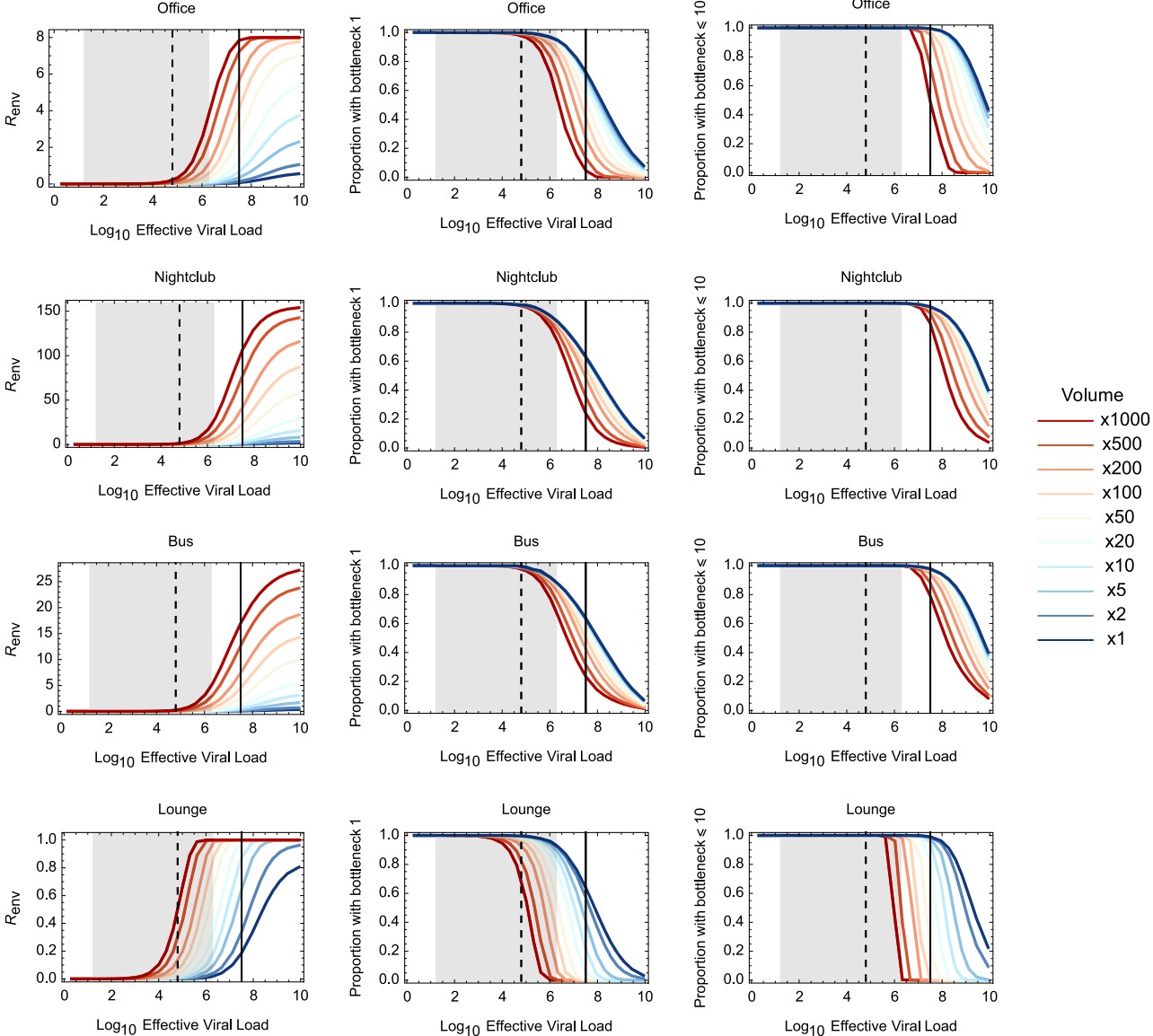

**Fig. 5 | Effect of increasing the volume of particles emitted upon the expected number of individuals infected and upon transmission bottleneck statistics.** Data show outputs from a model of coughing with increased emission volume. Bottleneck sizes were calculated from an ensemble of $10^6$ simulations for each combination of environment, volume emitted, and effective viral load. Where environments contained multiple uninfected people, increases in the volume of particles emitted led to substantial increases in the number of people infected. However, at all but the highest viral loads, most transmission bottlenecks still involved few viral particles.

While our model of particle spread captures the basic features of respiratory virus transmission, it still makes multiple simplifications. For example, our model neglects effects arising from convection currents caused by individuals in a room[41]. Effects such as these have the potential to generate non-monotonic levels of exposure with distance from an infected person, as particles are carried up and across the ceiling before falling to a height at which they can be breathed in. Our model also neglects the effects of local humidity on particle spread[42], as well as any detailed description of ventilation, such as the placement of windows, ceiling vents, and air conditioning units. Such effects are likely to have a distorting effect on the patterns of exposure we identify, potentially enhancing disparities in exposure. The potential effects of a highly skewed exposure distribution are represented by our simulated bus environment where the long and thin shape of the bus leads to extreme disparities in exposure (Supplementary Fig. 2). These disparities place an effective ceiling on $R_{env}$ (Fig. 5, Supplementary Fig. 5), with some people out of reach of emitted particles.

The transmission bottlenecks we inferred for the bus were not substantially distinct from those of other environments.

The simplifications involved in our model limit direct comparison with real-world scenarios. A recent publication suggested an infection risk of slightly under 10% for individuals in the most risky scenarios after 8 h of exposure[43], which compares to 18.7% or 74.9% in our default lounge models of coughing and continuous speech: As discussed in Supplementary Methods 1, our default parameters likely overestimate the effective viral load in a realistic situation. We note that a complete accounting for transmission would require an account of the precise distributions of emissions, viral load, and time-dependent proximity between individuals, alongside environmental parameters and a detailed description of human behaviour.

One area of uncertainty relevant to our model is the relationship between the raw numbers of viral particles contained in emitted material, the number of plaque-forming units (PFU) this represents, and the true effective viral load. The raw number of viruses in emitted

material, known as the viral load, follows a pattern of growth and then decay during the course of infection, which in SARS-CoV-2 infection reach a peak potentially of $10^{10}$ viruses per ml[44]. In our default model, we have used parameters from one study describing SARS-CoV-2 infection, which suggest a 3000-fold ratio between the raw number of viruses and the number of PFUs[35], further assuming a 1:1 ratio between PFUs and the effective viral load. We note considerable variation in the literature on this ratio. Experimental work has suggested a strain-dependent ratio between SARS-CoV-2 viral load and focus-forming units, a measure in some ways similar to PFU, of between $10^4$:1 and $10^6$:1[45], while a challenge study of SARS-CoV-2 infection estimated close to a $10^5$:1 ratio[46]. The relationship between PFU and the $TCID_{50}$, the dose needed to initiate infection in 50% of individuals, is a topic of some controversy, with a review of the subject identifying estimates spanning several orders of magnitude, from 1.26 to $7 \times 10^{6.25}$ PFU[47]. Modelling studies have attempted to estimate directly the ratio between raw and effective viral loads, with a study of super spreading events concluding on the basis of a Wells-Riley model suggesting a ratio of between 2000:1 and 300:1[48]. If we assume that the ability to form plaques under favourable experimental conditions is a necessary condition for a virus to cause infection in a host, and we allow for flexibility given the assumptions underlying modelling studies, our 3000:1 ratio is likely at the conservative end of the spectrum. Studies of viral load in other respiratory viruses show similar values to SARS-CoV-2. Data from infections with parainfluenza and respiratory syncytial virus show peak PFU levels around $10^6$/ml[49,50]. Studies of influenza show mixed results, with peak PFU values often between $10^4$ and $10^6$/ml but with occasional cases potentially reaching $10^9$ PFU/ml[51]. An interesting case is that of measles infection: The very high reported $R_0$ for this virus[52] makes this a potential case where transmission bottlenecks in unvaccinated individuals may be higher.

Uncertainty in the literature also exists around the precise distributions of particle sizes emitted via coughing, speaking and sneezing. While our method exploits experimental results, studies of these processes have historically used different methods and are not in perfect agreement. While we would not be confident about building a combined model of particle emission, combining, for example, speaking and sneezing, the finding that our basic result holds across such distinct models supports the robustness of our conclusions.

A final simplification in our model is the neglect of interactions between viruses, which could increase or decrease bottleneck sizes. Some interactions, such as those characterised by superinfection exclusion, are likely to reduce the number of cases of large transmission bottlenecks. In many cases of acute respiratory infection, a virus-founding infection leads to the rapid growth of viral particles[53], such that after a given amount of time, any subsequent infection will involve the addition of a tiny fraction of the current within-host population. This, alongside the triggering of innate host immune responses[54] and other cellular interactions[55], limits the window of time available for new viruses to infect a host. Other interactions between viruses involving cooperation have the potential to increase the proportion of bottlenecks involving multiple virions[56]. Where single virions contain incomplete functional genomes, more than one may be required for a cell to produce a complete genome[57,58]. A consideration of viruses with incomplete genomes would require a more nuanced definition of what is meant by a transmission bottleneck.

Despite its limitations, the generalisability of our model and the reproducibility of our result across a broad range of scenarios provide what we believe is a compelling explanation for past observations of tight transmission bottlenecks in respiratory virus transmission. Where the number of cases of infection in a scenario is limited, as represented by a moderate value of $R_{env}$, most people exposed to an infected person are not themselves infected, incurring an effective transmission bottleneck of zero. Where infectious particles spread through the air via diffusion, it is difficult to generate patterns of exposure that combine cases of non-infection with cases of infection that exclusively involve large bottlenecks. The mechanism of airborne respiratory virus transmission leads to tight transmission bottlenecks.

## Methods

### Wells-Riley model

The Wells-Riley model adopts a Poisson assumption about infection. Suppose that a person receives a level of exposure $E$, by which we mean that the expected number of viruses causing an infection is equal to $E$. Then the probability of an individual being infected is given by

$$P(\text{infection}) = 1 - e^{-E} \qquad (1)$$

While the derivation of $E$ can be complex, involving multiple individual and environmental factors, we here simply considered a range of possible values for this statistic.

We denote the population bottleneck at transmission by $N$. Given a Poisson model, the proportion of cases of infection initiated by a single viral particle is given by

$$P(N = 1) = \frac{E e^{-E}}{1 - e^{-E}} \qquad (2)$$

Similarly, the proportion of cases of infection initiated by ten or fewer viral particles is

$$P(N \le 10) = \frac{\sum_{k=1}^{10} \frac{E^k e^{-E}}{k!}}{1 - e^{-E}} \qquad (3)$$

These formulas were used to estimate transmission bottlenecks under different levels of exposure.

### Transmission mediated by the airborne spread of infectious particles

In order to estimate how exposures to a virus might vary in a given environment, we built a model of the airborne spread of viruses within a room. Considering the first instance SARS-CoV-2 infection, we modelled the behaviour of particles emitted by an infected individual with a cough, measuring the subsequent exposure of others in the room.

Considering a single coughing event, we estimated for each environment the exposure $E(\mathbf{x},r)$, describing the volume of emitted infectious material comprised of emitted particles of radius $r$ to which an uninfected person at position $\mathbf{x} = \{x,y\}$ would be exposed over a period of time. This expression was calculated by a process of summation: We generated an expression for the time-dependent exposure $E_C(\mathbf{x},r,t)$, occurring $t$ seconds after a single cough, and then summed over time and coughing events.

### Diffusion model

To calculate $E_C(\mathbf{x},r,t)$, we estimated the concentration of infectious material contained in particles of radius $r$ μm at $\mathbf{x} = \{x,y\}$ and time $t$ following a single emission event. This concentration is altered by the emission of infectious particles into the environment, the spread of particles through space, the loss of particles via evacuation and sedimentation, and the inactivation of viruses contained within particles. We write

$$\frac{\partial c}{\partial t} = \overbrace{I(x,t)}^{\text{Particle emission}} + \underbrace{K(L,Y)\left(\frac{\partial^2 c}{\partial x^2} + \frac{\partial^2 c}{\partial y^2}\right)}_{\text{Turbulent diffusion}} - \overbrace{B(r,c,\gamma)}^{\text{Evacuation}} - \underbrace{S(r,c)}_{\text{Sedimentation}} - \overbrace{D(c)}^{\text{Inactivation}}$$

$$(4)$$

We consider the parts of this equation in turn.

## Emission of infectious particles

Coughing leads to the emission of a distribution of particle sizes: This distribution has been studied via a range of experimental means[29,59]. Following this literature, we modelled particles emitted from a cough as following a lognormal distribution[30]. We simulated particles with radii $r \in \{1, 2, ..., 500\}$ µm, with particles being emitted in quantities proportional to the function

$$f(r) = Q(2r, u, s) - Q(2(r-1), u, s) \tag{5}$$

where $Q$ is the cumulative distribution function of the lognormal distribution

$$Q(d, u, s) = \frac{1}{2}\left[1 + \text{erf}\left(\frac{\ln d - u}{s\sqrt{2}}\right)\right] \tag{6}$$

with the parameters $u = 2.60269$ and $s = 0.693147$[60]. We assumed that the infected person coughed 10 times per hour at regular intervals[61].

The velocity of particles following a cough falls rapidly, within a fraction of a second[62]. We, therefore, described a cough as instantaneously creating a cloud of particles at mean radial distance of 20 cm from the infected person (standard deviation 5 cm) and with a spread angle 45%[63]. By default, the volume of liquid emitted from a cough was set to equal 38 pl[64]. Altering the initial mean radial distance of the cloud of particles had only a small impact on exposure levels (Supplementary Fig. S6).

We also investigated models of particle emission by coughing and sneezing. Descriptions of these models, and further details of the emission model, are provided in Supplementary Information.

## Evaporation

Emitted particles evaporate over time, the removal of liquid leaving behind a smaller solid particle with a radius of approximately one-quarter of that which was emitted[65–67]. This process occurs relatively quickly, with, for example, a droplet of size 20 µm evaporating in under a second[67] and a droplet of size 55 µm evaporating within an estimated 14.5 s[68]. Given the overall timescale of our model, we assumed that the process of evaporation is short, such that a particle of radius $r_0$ was, upon emission, instantaneously reduced to the new size $r = r_0/4$. In the following description, we refer to particles according to their radius at the time of emission.

## Turbulent diffusion

Once emitted, particles spread through the air via diffusion. Both Brownian motion and air turbulence potentially contribute to this, though at the size of particles we consider, it is likely that turbulent diffusion will dominate over Brownian motion[69]; our model therefore neglected the effects of Brownian motion.

The extent of turbulent diffusion depends upon how well a room is ventilated, with more frequent replacement of the air in a room, or a larger room, each requiring a higher mean rate of particle movement. We adopted a model based on the experimental measurement of air in a domestic environment[70]. This approach defined a characteristic length scale for a room by

$$L = \sqrt[3]{XYZ} = \sqrt[3]{V} \tag{7}$$

where $V$ is the volume of the room in m³. Our model then links $K$, the turbulent diffusion coefficient, to $L$, and $\gamma$, the number of changes of the air in a room per hour:

$$\frac{K}{L^2} = 0.52\gamma + 0.31 h^{-1} \tag{8}$$

Within our model, this becomes

$$K(L, \gamma) = (0.52\gamma + 0.31)L^2 \tag{9}$$

## Evacuation

We interpret the air change rate $\gamma$ using the cutoff radius theory presented by Bazant et al.[6]. Under this model, the evacuation rate is the same as the air replacement rate for droplets below a cutoff radius given by

$$r_c = \sqrt{\frac{9\gamma L \mu_a}{2g\Delta\rho}} \tag{10}$$

where $\mu_a$ is the dynamic viscosity of air, $g$ is the acceleration due to gravity, and $\Delta\rho$ is the difference in densities between water and air. Above the cutoff radius, the evacuation rate scales with $1/r^2$, with heavier particles being less subject to air movement. Where

$$r_* = \max\{r, r_c\} \tag{11}$$

we have

$$B(r, c) = -\gamma\left(\frac{r_c}{r_*}\right)^2 c \tag{12}$$

## Sedimentation

Emitted particles will be affected by gravity, with heavier particles falling to the floor more quickly than lighter particles, according to Stokes' Law. In our model we approximated this process by the simple removal of particles from the air over time. We followed a previous approach, which balanced diffusion and gravitational terms to approximate the time taken for a particle to fall to the ground[68]. We have

$$t_{sed} = \phi\frac{z_0}{r^2} \tag{13}$$

where $z_0$ is the initial height of the particle, $r$ is the particle radius, and $\phi$ is calculated as $0.85 \times 10^{-8}$ ms. From this, we derived the sedimentation term

$$S(r, c) = \frac{c}{t_{sed}} \tag{14}$$

The initial height of particles $z_0$ was defined according to whether individuals in an environment were standing or seated. We made the assumption that the floor absorbs particles, with no rebound being possible.

## Virus inactivation

Viruses within emitted particles are intrinsically unstable, such that the number of infectious particles in each droplet decays over time. An experimental study has suggested a half-life for SARS-CoV-2 of around 1.1 h[33]. The viral titre in each droplet is therefore given by

$$N(t) = N_0 e^{-\lambda t} \tag{15}$$

where $N_0$ is the initial number of particles in the droplet and $\lambda$ is the decay constant. To model a half-life of 1.1 h, we set $\lambda = 0.6301$ h⁻¹. We then have

$$D(c) = -\lambda c \tag{16}$$

## Solution of the diffusion equation

By default, we made the assumption that, upon hitting a wall, particles are absorbed, either impacting upon the wall due to electrostatic or inertial forces[71] or being caught in downward convection currents leading to their deposition on the floor[41]. By means of a sensitivity analysis, we also considered the case in which walls perfectly reflected particles. Under each of these conditions, the solution of Eq. (4) can be expressed analytically. Notes on the solution of the diffusion equation are provided in Supplementary Information.

## Calculating individual exposures

We generated values $c(\mathbf{x}, r, t)$ at time intervals of one second for a period of one hour following each emission event. The initial values $c(\mathbf{x}, r, 0)$ were scaled so that the total volume, summed across particle sizes, was equal to the desired volume of the emission. In order to calculate the total exposure of person i at the location $\mathbf{x_i} = \{x_i, y_i\}$, we generated values $c(\mathbf{x}, r, t)$ at positions in the square grid centred on $\mathbf{x_i}$, with dimension 40 cm, and containing points at resolution 2 cm, finding the mean value of c in this grid. The volume of the space represented by this box is $0.16Z$, where $Z$ is the height of the room so that we can calculate the density of viral particles of radius $r$ in the box. Given a parameter $A$, describing the rate of air inhalation by a person, we calculated the expected number of particles of radius $r$ inhaled within a 1 second interval at time $t$. Summing these values over times $t$, we obtained an expected number of particles of radius $r$ inhaled in a 1 hour period following a single emission event. Summing these values over multiple emission events, we obtained an expected number of particles of radius $r$ inhaled over the entire model period. We denote this number as $P_i(r)$.

For each uninfected individual in our model, we generated a Poisson random variable

$$n_{i,r} = \text{Poisson}(P_i(r)) \qquad (17)$$

describing the number of particles of radius $r$ inhaled by that person. This number was converted into a number of effective viruses: For each such particle, the expected number of effective viruses is given by

$$V_e(k_b, r) = \left(\frac{4}{3}\right) k_b \pi r^3 \qquad (18)$$

where $k_b$ is the effective viral load of particles at the point of emission. To calculate the effective number of viruses inhaled via particles of radius $r$, we calculated a second Poisson random variable

$$v_{i,r} = \text{Poisson}(n_{i,r} V_e(k_b, r)) \qquad (19)$$

The transmission bottleneck related to the person $i$ was finally calculated as the sum of these values:

$$N_i = \sum_r v_{i,r} \qquad (20)$$

Person $i$ was considered to have been infected if and only if $N_i > 0$. Statistics of bottlenecks were calculated across cases of infection.

For each scenario considered, we calculated $10^6$ independent simulations, generating $N_i$ for each individual in each simulation. Statistics were collated across simulations.

## Calculation of $k_b$

By default the effective viral load was calculated using an epidemiological model. Details are given in the Supplementary Information. A broad range of values of $k_b$ were considered.

## Inhalation

Our model assumes that the process of being exposed does not change the local level of exposure i.e. breathing in viruses does not significantly remove viruses from the air. We explore this assumption further in Supplementary Information.

## Environments

We modelled transmission within different environments, including an office, a bus, a nightclub, and a lounge. For each environment, our model was parameterised with the dimensions of the room in metres, $X$, $Y$, and $Z$, the number of uninfected people present, $n_e$, and their locations, the air replacement rate $\gamma$, the length of time for which we assumed people were in the environment $T$, the volume of air breathed in per minute by an individual, $A$, and the height at which particles were emitted, $z_0$. Parameters for each environment are shown in Table 1.

In Supplementary Information, we provide further notes on environmental parameters and a description of methods used to model variation in infectivity levels.

## Reporting summary

Further information on research design is available in the Nature Portfolio Reporting Summary linked to this article.

# Data availability

Data underlying the figures shown in this manuscript are available in the repository https://github.com/cjri/DiffusionCodeData/.

# Code availability

Code was used to generate data simulating and analysing levels of exposure in different environments. The code is available at https://github.com/cjri/DiffusionCode[72].

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

**Table 1 | Parameters defining each of the environments explored in our model**

| Environment | X (m) | Y (m) | Z (m) | T (h) | γ (h⁻¹) | $n_e$ | A (L/min) | $z_0$ (m) |
|---|---|---|---|---|---|---|---|---|
| Office | 10 | 3 | 10 | 8 | 3[73] | 8 | 10[74] | 1.1 |
| Nightclub | 10 | 6 | 15 | 4 | 25[75] | 162 | 40[74] | 1.6 |
| Bus | 2.4 | 2.4 | 12 | 1 | 10[76] | 50 | 12.5[77] | 1.1 |
| Living room | 5.5 | 2.7 | 3.7 | 8 | 1 | 1 | 6[74] | 1.1 |

Parameters describe the dimensions of the rooms X, Y, and Z, the time spent in the given environment, T, the number of air changes per hour, γ, the number of uninfected people in the environment, $n_e$, the volume of air breathed in per minute by uninfected individuals, A, here shown in units of litres, and the height at which particles were emitted, $z_0$.

8. Jones, B. et al. Modelling uncertainty in the relative risk of exposure to the SARS-CoV-2 virus by airborne aerosol transmission in well mixed indoor air. *Build. Environ.* **191**, 107617 (2021).

9. Buonanno, G., Morawska, L. & Stabile, L. Quantitative assessment of the risk of airborne transmission of SARS-CoV-2 infection: prospective and retrospective applications. *Environ. Int.* **145**, 106112 (2020).

10. Cheng, Y. et al. Face masks effectively limit the probability of SARS-CoV-2 transmission. *Science* **372**, 1439–1443 (2021).

11. Di Gilio, A. et al. $CO_2$ concentration monitoring inside educational buildings as a strategic tool to reduce the risk of Sars-CoV-2 airborne transmission. *Environ. Res.* **202**, 111560 (2021).

12. Bergstrom, C. T., McElhany, P. & Real, L. A. Transmission bottlenecks as determinants of virulence in rapidly evolving pathogens. *Proc. Natl Acad. Sci. USA* **96**, 5095–5100 (1999).

13. Morris, D. H. et al. Asynchrony between virus diversity and antibody selection limits influenza virus evolution. *eLife* **9**, e62105 (2020).

14. Leonard, A. S., Weissman, D. B., Greenbaum, B., Ghedin, E. & Koelle, K. Transmission bottleneck size estimation from pathogen deep-sequencing data, with an application to human Influenza A virus. *J. Virol.* **91**, 19 (2017).

15. McCrone, J. T. et al. Stochastic processes constrain the within and between host evolution of Influenza virus. *eLife* **7**, e35962 (2018).

16. Lythgoe, K. A. et al. SARS-CoV-2 within-host diversity and transmission. *Science* eabg0821 (2021) https://doi.org/10.1126/science.abg0821.

17. Bendall, E. E. et al. Rapid transmission and tight bottlenecks constrain the evolution of highly transmissible SARS-CoV-2 variants. *Nat. Commun.* **14**, 272 (2023).

18. Varble, A. et al. Influenza A virus transmission bottlenecks are defined by infection route and recipient host. *Cell Host Microbe* **16**, 691–700 (2014).

19. Sacristan, S., Diaz, M., Fraile, A. & Garcia-Arenal, F. Contact transmission of tobacco mosaic virus: a quantitative analysis of parameters relevant for virus evolution. *J. Virol.* **85**, 4974–4981 (2011).

20. Monsion, B., Froissart, R., Michalakis, Y. & Blanc, S. Large bottleneck size in cauliflower mosaic virus populations during host plant colonization. *PLoS Pathog.* **4**, e1000174 (2008).

21. Khiabanian, H., Emmett, K. J., Lee, A. & Rabadan, R. High-resolution genomic surveillance of 2014 ebolavirus using shared subclonal variants. *PLoS Curr.* https://doi.org/10.1371/currents.outbreaks.c7fd7946ba606c982668a96bcba43c90 (2015).

22. Sobel Leonard, A. et al. Deep sequencing of Influenza A virus from a human challenge study reveals a selective bottleneck and only limited intrahost genetic diversification. *J. Virol.* **90**, 11247–11258 (2016).

23. Lumby, C. K., Nene, N. R. & Illingworth, C. J. R. A novel framework for inferring parameters of transmission from viral sequence data. *PLoS Genet.* **14**, e1007718 (2018).

24. Xue, K. S. & Bloom, J. D. Reconciling disparate estimates of viral genetic diversity during human influenza infections. *Nat. Genet.* **51**, 1298–1301 (2019).

25. Martin, M. A. & Koelle, K. Reanalysis of deep-sequencing data from Austria points towards a small SARS-COV-2 transmission bottleneck on the order of one to three virions. *Sci. Transl. Med.* **13**, eabh1803 (2021).

26. Elena, S. F., Sanjuán, R., Bordería, A. V. & Turner, P. E. Transmission bottlenecks and the evolution of fitness in rapidly evolving RNA viruses. *Infect. Genet. Evol.* **1**, 41–48 (2001).

27. Tully, D. C. et al. Differences in the selection bottleneck between modes of sexual transmission influence the genetic composition of the HIV-1 founder virus. *PLoS Pathog.* **12**, e1005619 (2016).

28. Randall, K., Ewing, E. T., Marr, L. C., Jimenez, J. L. & Bourouiba, L. How did we get here: what are droplets and aerosols and how far do they go? A historical perspective on the transmission of respiratory infectious diseases. *Interface Focus* **11**, 20210049 (2021).

29. Duguid, J. P. The size and the duration of air-carriage of respiratory droplets and droplet-nuclei. *Epidemiol. Infect.* **44**, 471–479 (1946).

30. Chao, C. Y. H. et al. Characterization of expiration air jets and droplet size distributions immediately at the mouth opening. *J. Aerosol Sci.* **40**, 122–133 (2009).

31. Han, Z. Y., Weng, W. G. & Huang, Q. Y. Characterizations of particle size distribution of the droplets exhaled by sneeze. *J. R. Soc. Interface* **10**, 20130560 (2013).

32. Wei, J. & Li, Y. Airborne spread of infectious agents in the indoor environment. *Am. J. Infect. Control* **44**, S102–S108 (2016).

33. van Doremalen, N. et al. Aerosol and surface stability of SARS-CoV-2 as compared with SARS-CoV-1. *N. Engl. J. Med.* **382**, 1564–1567 (2020).

34. Hamner, L. et al. High SARS-CoV-2 attack rate following exposure at a choir practice—Skagit County, Washington, March 2020. *MMWR Morb. Mortal. Wkly. Rep.* **69**, 606–610 (2020).

35. Hakki, S. et al. Onset and window of SARS-CoV-2 infectiousness and temporal correlation with symptom onset: a prospective, longitudinal, community cohort study. *Lancet Respir. Med.* **10**, 1061–1073 (2022).

36. Delamater, P. L., Street, E. J., Leslie, T. F., Yang, Y. T. & Jacobsen, K. H. Complexity of the basic reproduction number ($R_0$). *Emerg. Infect. Dis.* **25**, 1–4 (2019).

37. Adam, D. C. et al. Clustering and superspreading potential of SARS-CoV-2 infections in Hong Kong. *Nat. Med.* **26**, 1714–1719 (2020).

38. Shen, Y. et al. Community outbreak investigation of SARS-CoV-2 transmission among bus riders in Eastern China. *JAMA Intern. Med.* **180**, 1665 (2020).

39. Asadi, S. et al. Aerosol emission and superemission during human speech increase with voice loudness. *Sci. Rep.* **9**, 2348 (2019).

40. Ghafari, M., Lumby, C. K., Weissman, D. B. & Illingworth, C. J. R. Inferring transmission bottleneck size from viral sequence data using a novel haplotype reconstruction method. *J. Virol.* **94**, 17 (2020).

41. Bhagat, R. K., Davies Wykes, M. S., Dalziel, S. B. & Linden, P. F. Effects of ventilation on the indoor spread of COVID-19. *J. Fluid Mech.* **903**, F1 (2020).

42. Oswin, H. P. et al. The dynamics of SARS-CoV-2 infectivity with changes in aerosol microenvironment. *Proc. Natl Acad. Sci. USA* **119**, e2200109119 (2022).

43. Ferretti, L. et al. Digital measurement of SARS-CoV-2 transmission risk from 7 million contacts. *Nature* https://doi.org/10.1038/s41586-023-06952-2 (2023).

44. Kleiboeker, S. et al. SARS-CoV-2 viral load assessment in respiratory samples. *J. Clin. Virol.* **129**, 104439 (2020).

45. Despres, H. W. et al. Measuring infectious SARS-CoV-2 in clinical samples reveals a higher viral titer:RNA ratio for Delta and Epsilon vs. Alpha variants. *Proc. Natl Acad. Sci. USA* **119**, e2116518119 (2022).

46. Killingley, B. et al. Safety, tolerability and viral kinetics during SARS-CoV-2 human challenge in young adults. *Nat. Med.* **28**, 1031–1041 (2022).

47. SeyedAlinaghi, S. et al. Minimum infective dose of severe acute respiratory syndrome coronavirus 2 based on the current evidence: a systematic review. *SAGE Open Med.* **10**, 205031212211150 (2022).

48. Prentiss, M., Chu, A. & Berggren, K. K. Finding the infectious dose for COVID-19 by applying an airborne-transmission model to superspreader events. *PLoS ONE* **17**, e0265816 (2022).

49. Pinky, L., Burke, C. W., Russell, C. J. & Smith, A. M. Quantifying dose-, strain-, and tissue-specific kinetics of parainfluenza virus infection. *PLoS Comput. Biol.* **17**, e1009299 (2021).

50. DeVincenzo, J. P. et al. Viral load drives disease in humans experimentally infected with respiratory syncytial virus. *Am. J. Respir. Crit. Care Med.* **182**, 1305–1314 (2010).

51. Hadjichrysanthou, C. et al. Understanding the within-host dynamics of influenza A virus: from theory to clinical implications. *J. R. Soc. Interface* **13**, 20160289 (2016).

52. Guerra, F. M. et al. The basic reproduction number (R 0) of measles: a systematic review. *Lancet Infect. Dis.* **17**, e420–e428 (2017).

53. Baccam, P., Beauchemin, C., Macken, C. A., Hayden, F. G. & Perelson, A. S. Kinetics of Influenza A virus infection in humans. *J. Virol.* **80**, 7590–7599 (2006).

54. Pawelek, K. A. et al. Modeling within-host dynamics of influenza virus infection including immune responses. *PLoS Comput. Biol.* **8**, e1002588 (2012).

55. Sims, A. et al. Superinfection exclusion creates spatially distinct influenza virus populations. *PLoS Biol.* **21**, e3001941 (2023).

56. Phipps, K. L. et al. Collective interactions augment influenza A virus replication in a host-dependent manner. *Nat. Microbiol.* **5**, 1158–1169 (2020).

57. Nee, S. The evolution of multicompartmental genomes in viruses. *J. Mol. Evol.* **25**, 277–281 (1985).

58. Jacobs, N. T. et al. Incomplete influenza A virus genomes occur frequently but are readily complemented during localized viral spread. *Nat. Commun.* **10**, 3526 (2019).

59. Morawska, L. Droplet fate in indoor environments, or can we prevent the spread of infection? *Indoor Air* **16**, 335–347 (2006).

60. Wang, Y., Xu, G. & Huang, Y.-W. Modeling the load of SARS-CoV-2 virus in human expelled particles during coughing and speaking. *PLoS ONE* **15**, e0241539 (2020).

61. Yousaf, N., Monteiro, W., Matos, S., Birring, S. B. & Pavord, I. D. Cough frequency in health and disease. *Eur. Respir. J.* **41**, 241–243 (2013).

62. Nishimura, H., Sakata, S. & Kaga, A. A new methodology for studying dynamics of aerosol particles in sneeze and cough using a digital high-vision, high-speed video system and vector analyses. *PLoS ONE* **8**, e80244 (2013).

63. Li, M. et al. Towards realistic simulations of human cough: effect of droplet emission duration and spread angle. *Int. J. Multiph. Flow* **147**, 103883 (2022).

64. Lindsley, W. G. et al. Quantity and size distribution of cough-generated aerosol particles produced by Influenza patients during and after illness. *J. Occup. Environ. Hyg.* **9**, 443–449 (2012).

65. Stadnytskyi, V., Bax, C. E., Bax, A. & Anfinrud, P. The airborne lifetime of small speech droplets and their potential importance in SARS-CoV-2 transmission. *Proc. Natl Acad. Sci. USA* **117**, 11875–11877 (2020).

66. Lieber, C., Melekidis, S., Koch, R. & Bauer, H.-J. Insights into the evaporation characteristics of saliva droplets and aerosols: levitation experiments and numerical modeling. *J. Aerosol Sci.* **154**, 105760 (2021).

67. Stiti, M., Castanet, G., Corber, A., Alden, M. & Berrocal, E. Transition from saliva droplets to solid aerosols in the context of COVID-19 spreading. *Environ. Res.* **204**, 112072 (2022).

68. Netz, R. R. Mechanisms of airborne infection via evaporating and sedimenting droplets produced by speaking. *J. Phys. Chem. B* **124**, 7093–7101 (2020).

69. Ounis, H. & Ahmadi, G. A comparison of Brownian and turbulent diffusion. *Aerosol Sci. Technol.* **13**, 47–53 (1990).

70. Cheng, K.-C. et al. Modeling exposure close to air pollution sources in naturally ventilated residences: association of turbulent diffusion coefficient with air change rate. *Environ. Sci. Technol.* **45**, 4016–4022 (2011).

71. Fletcher, L. A., Noakes, C. J., Sleigh, P. A., Beggs, C. B. & Shepherd, S. J. Air ion behavior in ventilated rooms. *Indoor Built Environ.* **17**, 173–182 (2008).

72. Sinclair, P., Zhao, L., Beggs, C. & Illingworth, C. The airborne transmission of viruses causes tight transmission bottlenecks. *Zenodo* https://doi.org/10.5281/zenodo.10953523 (2024).

73. Kolokotroni, M., Ren, X., Davies, M. & Mavrogianni, A. London's urban heat island: Impact on current and future energy consumption in office buildings. *Energy Build.* **47**, 302–311 (2012).

74. Pleil, J. D., Ariel Geer Wallace, M., Davis, M. D. & Matty, C. M. The physics of human breathing: flow, timing, volume, and pressure parameters for normal, on-demand, and ventilator respiration. *J. Breath Res.* **15**, 042002 (2021).

75. Engineering Toolbox. *Air Change Rates in Typical Rooms and Buildings* https://www.engineeringtoolbox.com/air-change-rate-room-d_867.html (2005).

76. Van Dyke, M. et al. Investigating dilution ventilation control strategies in a modern U.S. school bus in the context of the COVID-19 pandemic. *J. Occup. Environ. Hyg.* **19**, 271–280 (2022).

77. Zuurbier, M., Hoek, G., van den Hazel, P. & Brunekreef, B. Minute ventilation of cyclists, car and bus passengers: an experimental study. *Environ. Health* **8**, 48 (2009).

## Acknowledgements

This work was supported by funding from the UK Medical Research Council (MC_UU_12014 (P.S., C.I.), MC_UU_00034/6 (C.I.)).

## Author contributions

Conceptualisation: C.J.R.I. Methodology: P.S., L.Z., C.B.B., C.J.R.I. Software: P.S., C.J.R.I. Validation: P.S., C.J.R.I. Formal analysis: P.S., C.J.R.I. Investigation: P.S., C.J.R.I. Data curation: C.J.R.I. Writing—original draft: C.J.R.I. Writing—review and editing: P.S., L.Z., C.B.B., C.J.R.I. Visualisation: P.S., C.J.R.I. Supervision: C.J.R.I. Project administration: C.J.R.I. Funding acquisition: CJ.R.I.

## Competing interests

The authors declare no competing interests.
