## [Peer Review File · Nature Communications]

The airborne transmission of viruses causes tight transmission bottlenecksReviewers' comments:

Reviewer #1 (Remarks to the Author):

I thank the authors for a chance to review an interesting manuscript. I have the following comments

1) The introduction is somewhat oddly pitched, in that it discusses genetic info and then the model is simulation based. Also there is no mention of the quantitation microbial risk assessment literature that this work should sit within. Previous attempts at QMRA (source to infection) modelling should be referenced with the difference for this work shown.

2) The central term bottleneck is introduced without fully easing the reader into the rationale. Indeed when I have done QMRA modelling before we have discussed probabilities and infecting doses but bottleneck has been reserved for key stages in a process - such as say deposition in a host to receptor sites - where infection may be interrupted but the term is really used in QMRA literature. I have no major grievance at a new term but it would be good to justify it when the target journal has a wide readership (it may be ok to use in a discipline specific journal). In fact the author seem to be using bottleneck to refer to the infecting dose in general rather than say a threshold dose below which infection wouldnt be possible.

3) as well as framing in terms of QMRA it would be useful to frame in terms of wells riley modelling (that may be subject of citations - that I haven't checked in detail) but at present in manuscript it feels oblique at best.

4) the results are plausible but the main reason that domestic Renv is lower is the potential number of co-habitees/contacts. Which has less to do with environment and aerosol circulation than social mixing. This should be drawn out more in article if published. Particularly if ill people stay at home when ill due to symptoms/severity. Transmission is a intersection of virus, environment and behaviour and this study is focussed on environment and a little on virus but a lay reader may miss the lack of human behaviour impacting on results.

5) The static assumption is likely a necessary simplification but in reality the nightclub people may mix with friends and so a core cluster may be infected with random contacts more stochastic.

6) On symptoms the model depends on coughing and sneezing - these are some symptoms as cited in article. However, it is widely reported that people were asymptomatic (admittedly an ambiguous term) and more specifically had highest viral loads before symptom onset, so some statement in methods on chance of case developing a cough or sneeze and impact on results of case having no cough or sneeze would be useful. Even less clear is the relationship between viral load in host and symptoms/severity. It was noteworthy that 2 out of 18 cases in the human challenge study of Killingsley had higher viral loads than others.

7) There is a lack of framing with wider epidemic modelling literature for concepts such as R0 and infectious periods and the in host viral load literature.

Specific comments

1) Line 75: The statement "Infection occurs when an uninfected individual is exposed to infectious particles." is not true as this is simply exposure occurring. There are 'bottlenecks' such as respiration, deposition and immune response that may clear infectious material upon exposure.

2) Line 101: I appreciate this is a issue with nature style articles that results come before methods but this paragraph is a little confusing to parse, and I think the phrase effective viral load should be very clearly defined here for reader - viral load is normally used for the load in potential infector than infectees infecting dose.

3) line 118: why model a living room when people spend longer in bedroom asleep.

- 4) line 308, the choice of notation for evacuation E term in eqn is unhelpful given notation of exposure in line above
- 5) in general can some eqns be numbered to help reader follow the narrative arc in methods.
- 6) Line 326: can the authors justify log normal assumption or state this is a convenience assumption.
- 7) Line 371: How does the dispersion described in this section sit with the established literature on Gaussian puff or plume models?
- 8) line 447: Absorbption on surfaces is well cited here but what might happen is there was a degree of reflection - is this worth sensitivity analysis?
- 9) Line 490: Sentence doesn't make sense, is it meant to be framed as a question?
- 10) Line 550: Why 80 in the product - this is explained in text (I dont think, apologies if it is)
- 11) Line 578: The eqn defines I but this is not well described, this is then the cdf of the gamma distribution but this is not stated in text. Why might the infectious period be gamma? I know this is normally assumed in epidemic models on basis of sum of exponential processes but general reader may not know this. Indeed recent models of in host viral load suggest piecewise eponential or within host SIR type models that may offer other modelling options for sensitivity testing.
- 12) The methods are hard to follow and would be even harder for a lay reader and so some discussion to help reader on why length and time scales/dimensionality is needed would be useful

Reviewer #2 (Remarks to the Author):

Overall Comments:

Thank you for the opportunity to review your research. I found this work to be very interesting and highly novel, especially in how traditionally population-level measures of infectivity were related to estimates for individual infections. This is an important area of research that continues to grow, and I think this work represents a meaningful contribution. However, I think there is room for improvement in some of the modeling approaches and assumptions. Currently, I don't know that I "buy" the take home results (high proportion of infections caused by single viral particle), especially when it comes to differences across environments. I have listed major and minor comments for consideration below that I think will improve the robustness of the results of the study so that the significance of the results is in line with the innovativeness of the methodology.

Major Comments:

-The inhalation rates require a better explanation. While references are listed, these do not seem as comprehensive as other available sources (Table 1). For example, there are useful values from the U.S. Environmental Protection Agency Exposure Factors Handbook, Chapter 6, which include multiple sources for informing inhalation rates by age, sex, and activity level. For reference, age 21 to <31 have a mean inhalation rate of $4.2 \times 10^{-3} \text{ m}^3/\text{min}$. The high intensity inhalation rates are similar to the assumed value for the nightclub setting, but I am making the assumption that differences in inhalation rate across environment are due to expected differences in activity levels. If this is true, I don't see a rationale for a difference between an office, bus, and living room where I would expect sedentary/passive to light intensity. A better rationale should be provided here since this is an important exposure factor to consider that I would assume affects differences in bottlenecks by environment of exposure.

-A stochastic approach is taken with some aspects of the model but not with others. Because there

is variability and uncertainty in parameters like inhalation rate, air exchange rate, viral inactivation, I think these should also be addressed as stochastic variables. This would feel more methodologically and biologically consistent with approaches taken elsewhere in the model.

-Important references appear to be missing throughout. For example, definition for bottle neck (line 39), line 172 regarding distribution comparisons, etc. I recommend checking throughout and making sure references are easy to find that defend these important assumptions/concepts for the model.

-Where do the assumed air exchange rates for the different environments come from? Was natural ventilation and/or mechanical ventilation considered?

-The "propensity for infections" (line 570) feels very unclear right now. Are there units for this or a statement to help with interpretability? For example, in line 593, "a value of phi office much greater than one seems unlikely" doesn't have much meaning to me since I don't know what 1 is relative to or how this compares to other environments.

-It's unclear whether this work hinges on the idea that there are a required number of viral particles to initiate an infection, a minimal infectious dose. Is it more accurate to say that this work assumes any amount of viral particles poses a risk but that, in some cases, the dose amount that initiates infection will vary with consequences for genetic diversity of the viral particles initiating infection? Because the topic of minimal infectious dose is of such heated debate in some circles, especially quantitative microbial risk assessment communities, I think this could be worth clarifying in the introduction.

Minor Comments:

-In my experience, a more commonly used term in place of viral "degradation" would be viral "inactivation." This is prevalent in viral exposure modeling research and I think will be more interpretable by those audiences.

-In Table 1, the references look like exponents, and I found the values very surprising at first. Ha! Perhaps having reference columns per parameter would be useful here for clarity.

-Line 13: "found an infection" – like initiate an infection? The word "found" threw me off here.

-Line 59: "Ventilation removes particles from the air" – or dilutes, depending on the ventilation system. I think this should be clarified since ventilation has been such a contentious and poorly understood topic for some during COVID-19.

-Line 118 – It's unclear what this parameter means until much further in the Methods at the end of the paper. I recommend a more comprehensive definition at its first mention in the main text.

-Lines 124-130 read in a confusing way. It's unclear whether these findings are just due to the model dynamics or if they are anticipated to actually reflect reality or both.

-Line 589, what does "not accessible to our model" mean? In the same line as this, what does "required number of infections could not be obtained for any effective load" in figure 3 mean? Giving more detail about the computational limitation here would be useful for modelers.

-Figure 3 – labels for the color heatmap bars would be useful so that not all of the figure title/caption text has to be read to interpret.

Reviewer #3 (Remarks to the Author):

It is a useful attempt to derive the bottleneck size for a respiratory virus using an alternative approach than the commonly used genome sequencing data. I am glad that the authors tried to do it.

However, there appear several major issues with the mathematical models used in the current study and the choice of the studied settings.

I also have some concerns over the conclusion that it is airborne transmission that causes the tight transmission bottleneck, which is probably not fully supported by the data so far in the paper. I will probably focus on the method aspects. Hope that these questions may help the authors to refine their model.

First, the effective viral load k_b is probably the most important and crucial parameter. In this study, its value is calculated to match a specific R value (Line 552-553). This is right. The authors probably assumed that everything in their model was correct, so that the model could back calculate the number concentration of infectious virus particles (virions) in the expired fluid. There is no validation of the back calculated number concentration of infectious viruses in the expired fluid (to my knowledge, such data also does not exist for the general population).

It is also known that expired droplets of varying sizes may contain various number of virions (viruses or particles as in Line 528-554). It appears that the possibility of droplets containing multiple infectious viruses is not dealt with in this paper.

The authors could have used the existing viral load data in the literature, then the crucial dose-response parameter is needed.

I do not fully understand the rationale behind the choice of R_{office} value – an $T_{office} = 0.02675$ was chosen. As $R_0=5$ is chosen, so that R_{office} is around 0.1. I do not understand about the rationale of this approach – do you mean that you have considered the situation in a “typical” office that you considered, R_{office} (as used in equation Line 550) equals to 0.02675×0.5 ? This does not seem right to me.

By the way, for the ancestral strain of SARS-CoV-2, I recall that most cited R_0 is about 2-2.5 (a systematic review of literature is needed), not 5 (Line 588)

Second, I have major concerns about the use of a 1D diffusion model (Line 308). There is no validation of the predicted spatial expired particle (droplets) with any experimental data. Only the final bottleneck size data are presented. It is almost impossible to judge or reproduce the model.

The 1D means that author needs to choose a dimension x – (width or length of the room, presumably in front of the index case, what do you do with the space behind the index case), and then consider a radius of exposure x (Line 490) for a susceptible individual at a location (Line 474). The authors were also able to consider the boundary conditions at walls (again I am not sure how the four walls can be considered in a 1D simulation, except if you do the calculation in all directions (360 degree around the infected, I also guessed that the authors did this).

This might be a very crude assumption as the turbulent diffusion approach is not good at modelling the expired puffed jets.

Third, many model decisions and parameters used are not valid to me. The assumption that the infectious virus particles are only released by cough is problematic. It is now known that breathing also emission particles. I cannot imagine how a cough (10 times per unit time) without mouth covering be possible in some of the modelled environment?

The cough droplets can also be bimodal, not a single mode lognormal (Line 319).

I also do not know why an artificial 80 micron was assumed as the largest droplets, as larger ones were found. Patients do not sneeze – not sure why sneeze is also considered (Line 346).

What was the rationale of choosing final size after evaporation as $r_0/4$ (Line 369)?

Would the simple diffusion measurement in ref 50 be applicable to all four venues considered in this study?

I am suspicious of the typical air change rates (Table 1) including using the data from Engineering Toolbox (some data without reference there). One big question to ask – if these are typical air change rates, would those outbreaks also occurred in settings with these air change rates, if not would your estimate of k_b value with $R_0=5$ be suspicious?

The inhalation/exhalation rate is 10 in office, and 12.5 on bus – I would have thought that passengers would sit quietly and in an office environment, workers have a greater met value.

I stop here and I may suggest authors to consider a few real outbreak settings where detailed ventilation data is available.

Reviewers' comments:

We thank the reviewers for their comments on our manuscript, which have been extremely helpful in shaping a near-complete rewrite of this work.

Most significantly, we have rebuilt from scratch our model of particle spread: We now describe the spread of particles in an explicitly-modelled two-dimensional environment, with implicit terms describing the movement of particles in a vertical direction (e.g. in sedimentation, and calculating concentration per unit volume). We use this framework to model room boundaries in an explicit manner.

This change has been accompanied with a near-complete rewrite of the manuscript: For example our new model provides greater potential to explore the sensitivity of the model to a variety of parameters suggested by the reviewers. Our key result, that in the majority of circumstances, the majority of transmission bottlenecks are likely to be narrow, is unchanged, although we now elaborate on this: Two circumstances exist where this is not correct. Firstly, extremely high concentrations of viruses in emitted material could create a situation whereby the inhalation of a single emitted particle transmits sufficient viral material to start an infection with multiple viruses. Secondly, extreme levels of viral emission can create a scenario in which most cases of infection are initiated by multiple viruses. We discuss each circumstance, and show given biological data on viral infection that each represents an outlier to the normal pattern of viral infection.

A detailed response to comments received is provided below. Our response is in red text with quotes from the revised manuscript in blue.

Reviewer #1 (Remarks to the Author):

I thank the authors for a chance to review an interesting manuscript. I have the following comments

1) The introduction is somewhat oddly pitched, in that it discusses genetic info and then the model is simulation based. Also there is no mention of the quantitation microbial risk assessment literature that this work should sit within. Previous attempts at QMRA (source to infection) modelling should be referenced with the difference for this work shown.

We have substantially revised the introduction, but note an important difference between our topic and the general focus of QMRA. What we believe is different about our approach relative to QMRA is that QMRA considers the risk of a person being infected by an infectious agent, while we are concerned with the question of how many viruses initiate infection, given that a person is infected.

The primary target of this manuscript is the evolutionary biology community, specifically that part of it concerned with infectious disease and transmission. The subject of transmission bottlenecks has been an important one in recent years, with multiple high-impact papers on this subject seeking to use viral genome sequence data to infer the number of viruses responsible for initiating infection in a novel host. We address this evolutionary question in a profoundly novel way, using modelling techniques as would perhaps more commonly be used in QMRA. This approach has a key advantage over genomic analyses. Genomic analyses of transmission bottlenecks rely upon the collection of data which may be misleading (e.g. in the precise determination of who infected who), which requires substantial effort and resources to obtain, and which ultimately describes only specific circumstances: One virus in one context. Our approach provides a more general solution,

noting that the process of viral emission and spread common to respiratory virus transmission has consequences for the transmission bottleneck.

We have softened the introduction for readers from outside of the evolutionary community. We now note the background of studies describing the mechanism of transmission, and evaluating the risk of transmission, before going on to our main point.

We are aware that our work is attempting to span multiple disciplines, and that we are limited in the number of references we can include in the manuscript. If there are specific references we are very clearly missing we are open to suggestions.

2) The central term bottleneck is introduced without fully easing the reader into the rationale. Indeed when I have done QMRA modelling before we have discussed probabilities and infecting doses but bottleneck has been reserved for key stages in a process - such as say deposition in a host to receptor sites - where infection may be interrupted but the term is really used in QMRA literature. I have no major grievance at a new term but it would be good to justify it when the target journal has a wide readership (it may be ok to use in a discipline specific journal). In fact the author seem to be using bottleneck to refer to the infecting dose in general rather than say a threshold dose below which infection wouldnt be possible.

Indeed, multiple bottlenecks are relevant to the virus lifecycle. There will be a bottleneck in terms of which viruses are emitted from the host, another at the point where viruses initiate an infection, and further bottlenecks within the host arising from the spatial distribution of viruses within the airway.

The term transmission bottleneck is well understood within the evolutionary literature, but we realise a better explanation is needed. We have adopted the following:

Given that a person was infected, how many viruses initiate that infection, so as to become the ancestors of all of the viruses produced during infection? This number of viruses, denoted the transmission bottleneck¹², has an important influence upon virus evolution.

This explanation leans on concepts from evolutionary biology, such as that of the most recent common ancestor, from which all individuals in a population are descended. We are happy to provide further clarification if required. We are indeed referring to the number of viruses which in practice initiate infection, rather than a threshold dose (e.g. of emitted particles) below which infection wouldn't be possible.

3) as well as framing in terms of QMRA it would be useful to frame in terms of wells riley modelling (that may be subject of citations - that I haven't checked in detail) but at present in manuscript it feels oblique at best.

We have thought about how to talk about Wells Riley modelling. The solution we arrived at was to describe a (very simple) Wells Riley model as a precursor to our full model. We describe results for a case in which every individual present receives an equal level of exposure before elaborating on how levels of exposure might vary within an environment.

4) the results are plausible but the main reason that domestic Renv is lower is the potential number of co-habitees/contacts. Which has less to do with environment and aerosol circulation than social mixing. This should be drawn out more in article if published. Particularly if ill people stay at home when ill due to symptoms/severity. Transmission is a intersection of virus, environment and behaviour and this study is focussed on environment and a little on virus but a lay reader may miss the lack of human behaviour impacting on

results.

We agree. The maximum value of R_{env} is equal to the number of people in a given environment. The lounge environment is intended to maximise the exposure of a single individual without having other uninfected people further away, so that this environment has a maximum R_{env} of 1.

The value of R_{env} has mixed importance in our model. On the one hand, it is a secondary output of the model, in so far as the output we are most concerned about is the distribution of transmission bottleneck sizes. R_{env} becomes important where we consider the second case we find where bottleneck sizes are generally large: In this case the values of R_{env} become very high. In theory we could convert everything into a percentage of people present who are infected, which would provide a normalisation for this statistic, but we chose to keep the raw number for easy comparison to R_0 : A case where over 100 people are infected in the nightclub has more immediate impact than one in which 60% of those present are infected, even though the two statistics are equivalent.

We have added language alluding to R_{env} having a theoretical maximum, for example:

[In the office] R_{env} in the office was close to the potential maximum value of 8,

... R_{env} values for the bus and nightclub environments were more than 16 and 100 respectively, reflecting high proportions of the numbers of individuals present being infected

The point about behaviour is really helpful, and we have added a comment to this extent in the discussion.

... a complete accounting for transmission would require an account of the precise distributions of emissions, viral load, and time-dependent proximity, in addition to environmental parameters and human behaviour.

5) The static assumption is likely a necessary simplification but in reality the nightclub people may mix with friends and so a core cluster may be infected with random contacts more stochastic.

Our default model is that people are static, which in some sense represents an extreme scenario of lack of movement. The Wells Riley model we include details what is in some ways the opposite extreme: If everyone in the room moved extremely quickly (by which we mean at unrealistically high speeds), this would approach a case in which everyone received the same level of exposure.

We have added results for a version of the nightclub scenario in which people randomly swap locations every five minutes: This produces a more even distribution of exposures, but does not result in a large change in the inferred bottleneck sizes.

6) On symptoms the model depends on coughing and sneezing - these are some symptoms as cited in article. However, it is widely reported that people were asymptomatic (admittedly an ambiguous term) and more specifically had highest viral loads before symptom onset, so some statement in methods on chance of case developing a cough or sneeze and impact on results of case having no cough or sneeze would be useful. Even less clear is the relationship between viral load in host and symptoms/severity. It was noteworthy that 2 out of 18 cases in the human challenge study of Killingsley had higher viral loads than others.

We have added to our manuscript a model of particle emission via speech, which captures the case of asymptomatic transmission. We compare this to our coughing model and show that they are similar except for a scalar parameter (i.e. the exposures received by individuals are highly correlated between our coughing and speaking models).

The relationship between viral load in the host and symptoms/severity is a complex one. Rather than describe the symptoms of an infection and then derive a combined model (which would depend upon the behaviour of the infected host), we characterise particle emissions in terms of:

- i) A volume of particles emitted. The default amount is set based upon the available literature.
- ii) A distribution of the sizes of particles that in total make up this volume, again set according to the kind of emission and based upon the available literature.
- iii) An effective viral load.

We consider uncertainty in each of these parameters and compare different distributions of particle sizes according to different emission types. Our result is derived from the overall picture given by these results. We think that the variation we consider accounts for the potential variation in viral load and symptoms/severity, albeit in an indirect manner.

7) There is a lack of framing with wider epidemic modelling literature for concepts such as R_0 and infectious periods and the in host viral load literature

We have clarified our definition of R_0 and provided a citation to a description of this statistic.

R_0 describes the expected total number of infections caused by an infected individual during the entire course of an infection, in the absence of population immunity.

We have altered our approach to the use of epidemiology to infer an effective viral load, in effect reducing the importance of R_0 in our model. We do use an estimate of R_0 to set the terms of our default parameters (details are now provided in Supplementary Text) but in this revised version manuscript we evaluate over a range of effective viral loads in a systematic manner. Similarly, the infectious period is mentioned, but is only really important in setting the default value of the effective viral load: Our results are derived from a systematic evaluation of model outputs across a broad range of effective viral loads.

Specific comments

1) Line 75: The statement "Infection occurs when an uninfected individual is exposed to infectious particles." is not true as this is simply exposure occurring. There are 'bottlenecks' such as respiration, deposition and immune response that may clear infectious material upon exposure.

We have clarified the language in our manuscript

"Infection occurs if and only if a person is infected by one or more viruses"

2) Line 101: I appreciate this is a issue with nature style articles that results come before methods but this paragraph is a little confusing to parse, and I think the phrase effective viral load should be very clearly defined here for reader - viral load is normally used for the load in

potential infector than infectees infecting dose.

We agree, and have substantially rewritten the introduction, for example to provide a clearer guide to the effective viral load.

Calculated physical exposures were converted into viral exposures (Figure 2B) using an estimate of a statistic we denote as the effective viral load. The effective viral load in our model describes the number of viruses per ml of emitted material that are expected to overcome the various barriers, whether physical or immunological, to initiate infection in the recipient.

3) line 118: why model a living room when people spend longer in bedroom asleep.

Environments were chosen to differ from one another in terms of their physical environments. The lounge represents a case of extended proximity between two individuals in an environment with poor ventilation. Although we could model a bedroom this would likely involve some complicated effects: If a person is lying with their face sideways on a bed, and then coughs, the interaction between the emitted particles and the bed itself would need to be considered.

4) line 308, the choice of notation for evacuation E term in eqn is unhelpful given notation of exposure in line above

We have chosen an alternative notation. Evacuation is now denoted using the letter B.

5) in general can some eqns be numbered to help reader follow the narrative arc in methods.

Yes, we have numbered equations throughout.

6) Line 326: can the authors justify log normal assumption or state this is a convenience assumption.

The distributions and parameters we used to describe the sizes of particle emissions were taken from experimental studies. These studies commonly use a lognormal distribution to summarise the data they describe.

The underlying reason why emitted particles should follow a lognormal distribution is beyond the scope of this study, but we found the following reference of interest: (Andersson, Mechanisms for log normal concentration distributions in the environment, Sci. Rep. 11, 16418, 2021)

7) Line 371: How does the dispersion described in this section sit with the established literature on Gaussian puff or plume models?

Our model differs from Gaussian puff and plume models in so far as, other than in the earliest phase of emission, the diffusion of particles in our model does not have a specific directionality. However, there are some similarities.

In our understanding Gaussian puff and plume models describe the diffusion of particles emitted from a point source in the presence of a specific external influence (e.g. a prevailing wind). In the direction perpendicular to this influence there is a process of Gaussian diffusion, such that the cross-section of the particle density is a normal distribution.

In terms of its dynamics our model is similar to this, but without any systematic external influence: A parallel arises if the wind speed is equal to zero. Although within a room there

will be specific air currents, we approximate these by making them omnidirectional. In essence we begin with a room having a specific size, in which there are a specified number of air changes per hour. The need for the air to be replaced requires air movement: The more air changes per hour and the bigger the room, the faster these movements will be. Under our model, these air currents are approximated as flowing in every direction at once. This creates a process of Gaussian diffusion. The assumption creates behaviour equivalent to that of the Gaussian plume in the direction perpendicular to that of the prevailing wind, but in our case this behaviour happens in both of the two dimensions we model.

Our model has some similarities with a Gaussian puff model, in the sense that we consider a cough or sneeze (or speaking event, representing one second's worth of emissions, but this is less of a good analogy) similarly to a puff. The emissions from a cough represent a discrete set of particles, which appear and then are subject to diffusion.

Parallels and differences exist between the initial state of our model and puff/plume models. In so far as we are modelling exposures over times on the scale of hours, we assume that the emissions follow something of a plume model, albeit one in which particles are emitted very fast, and then slow down very quickly: In our model a cough instantaneously creates a set of particles which are distributed in space.

The shape of our initial state is constructed via two successive approximations. The first approximation is similar to that of a Gaussian plume: Particles are emitted in something of a two-dimensional cone-shaped distribution. A technical difference between a plume and our first-approximation initial distribution is that our distribution is constructed using polar coordinates. Where a Gaussian plume emitted in the x-direction has a cross-section in the y-direction that is normally distributed, our emission model describes a plume emitted in the radial direction (i.e. in a direction of constant angle), that at any given radius is normally distributed in the angular direction. We derived this approximation based upon reports in the literature of coughs and sneezes having a specific 'spread angle'. Expressing a distribution in polar coordinates we can specify that 95% of emitted particles fall within the specified spread angle immediately following particle emission.

Having in a first approximation created an initial distribution of particles, our second approximation involves the fitting of an analytical function to the particle density. This is necessary for our calculation of particle dynamics, but creates a distortion to the distribution: We can capture only the mean position of emitted particles, and the variance of this distribution, characterised by a single parameter. The effect of this approximation is to squash the shape of the emitted distribution: For a sneeze, where the spread angle is only 15 degrees, particles become more concentrated in a region of space that is shorter and wider than that described by the polar coordinates model. For a cough, where the spread angle is 45 degrees, this distortion is lessened.

Putting the above discussion in the context of our manuscript as a whole, the analytical approximation we make in describing particle diffusion, required for computational feasibility, means that we are likely not very accurate in the way that the initial distribution of particles (particularly for a sneeze, less so for a cough) is represented. However, in the different environments we model, most people are far enough away from the source of emissions that any error in this component of the model will be washed out by subsequent particle diffusion.

We have carried out a sensitivity analysis to the initial location of the emitted particles in a cough (Supplementary Figure S13).

8) line 447: Absorption on surfaces is well cited here but what might happen is there was a degree of reflection - is this worth sensitivity analysis?

We agree. This was part of the motivation for expressing our model explicitly within a two-dimensional framework. Our analytical model allows for either the absorption of particles by the walls of the room, or for perfect reflection. By way of a sensitivity analysis, we compare outputs from the two models. Results are shown in Supplementary Figure S10.

9) Line 490: Sentence doesn't make sense, is it meant to be framed as a question?

There was a misplaced capital letter in the previous manuscript. We have altered the text for clarity.

10) Line 550: Why 80 in the product - this is explained in text (I don't think, apologies if it is)

The 80 was there because our model considered particles with radii between 1 and 80 μm . In response to another reviewer we have increased the maximum particle size that we consider. This text has been moved to supplementary information. We have amended the formula to show simply the product over all r . Also we now model particle sizes up to 500 μm for a cough.

11) Line 578: The eqn defines I but this is not well described, this is then the cdf of the gamma distribution but this is not stated in text. Why might the infectious period be gamma? I know this is normally assumed in epidemic models on basis of sum of exponential processes but general reader may not know this. Indeed recent models of in host viral load suggest piecewise exponential or within host SIR type models that may offer other modelling options for sensitivity testing.

This section of the Methods was not well explained. The gamma distribution and its parameters come from a published estimate in the literature. The reason why a gamma distribution is used to describe infectivity data is likely beyond the scope of our manuscript. Our use of the distribution is limited to using it to estimate an infectious period of 18 days, which itself is only used to set the default effective viral load.

We have moved the details of this section to Supplementary Information.

12) The methods are hard to follow and would be even harder for a lay reader and so some discussion to help reader on why length and time scales/dimensionality is needed would be useful

We have substantially rewritten the Methods section and believe it is clearer now. Less important sections have been moved to Supplementary Text. We are not entirely sure we understand the comment from the reviewer, but the purpose of our spatial model is to look at differences in exposure levels.

“In order to estimate how exposures to a virus might vary in a given environment...”

Reviewer #2 (Remarks to the Author):

Overall Comments:

Thank you for the opportunity to review your research. I found this work to be very interesting and highly novel, especially in how traditionally population-level measures of infectivity were related to estimates for individual infections. This is an important area of research that continues to grow, and I think this work represents a meaningful contribution. However, I think there is room for improvement in some of the modeling approaches and

assumptions. Currently, I don't know that I "buy" the take home results (high proportion of infections caused by single viral particle), especially when it comes to differences across environments. I have listed major and minor comments for consideration below that I think will improve the robustness of the results of the study so that the significance of the results is in line with the innovativeness of the methodology.

We thank the reviewer for their positive comments about our work. Our contention is that the result we obtain is robust to realistic changes in our model parameters.

Major Comments:

-The inhalation rates require a better explanation. While references are listed, these do not seem as comprehensive as other available sources (Table 1). For example, there are useful values from the U.S. Environmental Protection Agency Exposure Factors Handbook, Chapter 6, which include multiple sources for informing inhalation rates by age, sex, and activity level. For reference, age 21 to <31 have a mean inhalation rate of $4.2 \times 10^{-3} \text{ m}^3/\text{min}$. The high intensity inhalation rates are similar to the assumed value for the nightclub setting, but I am making the assumption that differences in inhalation rate across environment are due to expected differences in activity levels. If this is true, I don't see a rationale for a difference between an office, bus, and living room where I would expect sedentary/passive to light intensity. A better rationale should be provided here since this is an important exposure factor to consider that I would assume affects differences in bottlenecks by environment of exposure.

Inhalation rates don't make a huge difference to the bottleneck sizes, partly because they have a linear effect on the exposure, whereas differences in viral load shift on a log scale. The scale of differences between exposure levels is illustrated by the new Supplementary Figure S1. In order for the majority of exposures to lead to a bottleneck of size 10 or more, the level of exposure has to be shifted, in most cases, by multiple orders of magnitude relative to the default model. This is illustrated by a further new addition to the manuscript, where we arbitrarily scale the level of emissions from the basis of our default model, exploring a 1000-fold increase in emissions. If, in the inhalation rates we are out by a factor of two, this is not entirely without importance, but is well within the range that we explore.

Concerning the inhalation rates themselves, the inhalation rates were estimated from the published papers in the Table. The inhalation rates were derived from the published papers detailed in the Table. Three were derived from Pleil et al.'s 2021 paper in the Journal of Breath Research. The living room value was taken from the nominal at-rest rate of inhalation, assuming purely sedentary behaviour while the nightclub value was taken from a value describing 'moderate exercise'. The value for the office was estimated as a value slightly higher than that for resting i.e. 10 versus 6 litres per minute. The value for a bus was taken from the study of Zurbier et al, who estimated a value based upon heart monitor data of study participants on buses.

We acknowledge that these values are estimates, and may be imperfect, although we are reassured that the high intensity values in the handbook you mention are similar to the value we use for the nightclub.

A stochastic approach is taken with some aspects of the model but not with others. Because there is variability and uncertainty in parameters like inhalation rate, air exchange rate, viral inactivation, I think these should also be addressed as stochastic variables. This would feel more methodologically and biologically consistent with approaches taken elsewhere in the model.

We model the direct processes of infection as being stochastic, as alternative approaches would not work. For example, some people in our model have a level of exposure giving them a 1% chance of being infected: Under a stochastic model, they get infected 1% of the time, which seems realistic: We cannot have people being infected by fractions of viruses.

Rather than making parameters stochastic, we have explored the sensitivity of our model to some of these parameters, on the basis that i) making small stochastic changes to the air exchange rate will have a substantially lesser effect than systematically increasing or decreasing this parameter and ii) making some parameters stochastic would require an order of magnitude more calculation. Changing the inhalation rate or magnitude of emissions is straightforward, but changing the air exchange rate requires a full rerun of the part of the code that generates physical exposure values.

As noted above, we have effectively carried out a sensitivity analysis to changes in the inhalation rate by modifying the volume of particles emitted: The two in our model are equivalent (Note that we neglect the effect of inhalation upon the density of particles in the room. An evaluation showing that this has only a small effect upon the results is shown in Supplementary Figure S14). Changing the air exchange rate has some interesting consequences: We explicitly carry out a sensitivity analysis to this, shown in Supplementary Figure S7. Changing the viral inactivation rate makes less difference to the model output: Results are shown in Supplementary Figure S8.

In so far as we run 10^6 simulations with any given set of parameters, any stochasticity in parameters will be well-approximated by systematically describing a distribution over a parameter. We have taken this approach to perform calculations with variation in the extent of infectivity of the infected person in the room: The extent of infectivity acts as something of a proxy for variation in other parameters in the model, generally increasing or decreasing the exposure of individuals present in each of the simulations performed.

-Important references appear to be missing throughout. For example, definition for bottle neck (line 39), line 172 regarding distribution comparisons, etc. I recommend checking throughout and making sure references are easy to find that defend these important assumptions/concepts for the model.

We have gone through the text and added references where appropriate. The concept of a transmission bottleneck is derived from the more general concept of a population bottleneck: We have added a reference for this, and the appropriate reference for comparing distributions.

-Where do the assumed air exchange rates for the different environments come from? Was natural ventilation and/or mechanical ventilation considered?

The air exchange rates come from different parts of the literature; citations are given where appropriate. For example, the rate for an office comes from a study of energy usage in office buildings, while that for a bus comes from a study of COVID prevention in school buses in the US.

The mechanism of ventilation is not explicitly considered within our model. As we note in the discussion our model includes multiple simplifications, for example not modelling the placements of windows, ceiling vents and air conditioning units. Rather, ventilation is modelled as a mean effect, removing airborne particles from the room.

We believe that a more accurate model of ventilation would describe a distortion of the level of exposure at distinct points in a room, which would be time-dependent. For example, a

window would provide both an inflow and outflow of air, affected by the strength and direction of the wind outside. A full computational fluid dynamics model of a room would also include convection currents, and should likely consider movements of air caused by the movements of people. Such a model would require extensive calculation, yet the results would be very specific to the particular circumstances of that model. Our approach here is to use a more approximate model, applied to a set of distinct circumstances, from which we derive a general set of results.

-The “propensity for infections” (line 570) feels very unclear right now. Are there units for this or a statement to help with interpretability? For example, in line 593, “a value of ϕ office much greater than one seems unlikely” doesn’t have much meaning to me since I don’t know what 1 is relative to or how this compares to other environments.

Coming up with a precise value for the propensity of infections in an environment is difficult. The basic idea is simple: The parameter R_0 describes the expected number of naïve individuals an infected person will infect. Environment plays a role in the number of people an infected person will infect. The value R_0 thus makes an explicit assumption: This is the number of infections expected to occur over some mean ensemble of environments. We can arbitrarily assign the value 1 to be the mean propensity of these (unknown) environments to cause infection. It is clear that some environments are more likely than others to be places where infection occurs. However, describing what the ‘mean environment’ in which a person transmits infection is very hard.

Our revised text relegates ϕ_{env} to Supplementary Information: We use it only to set the default parameters for our model.

The term ϕ_{env} is unitless, as it describes a ratio of the relative propensity of a specific environment versus that of the “mean environment”.

-It’s unclear whether this work hinges on the idea that there are a required number of viral particles to initiate an infection, a minimal infectious dose. Is it more accurate to say that this work assumes any amount of viral particles poses a risk but that, in some cases, the dose amount that initiates infection will vary with consequences for genetic diversity of the viral particles initiating infection? Because the topic of minimal infectious dose is of such heated debate in some circles, especially quantitative microbial risk assessment communities, I think this could be worth clarifying in the introduction.

As we understand the question, the reviewer is referring to an assumption that a minimum exposure to viruses is required to initiate an infection. Our model does not make this assumption, except to note that exposure to at least one virus is necessary for infection to occur.

What the reviewer says is basically correct. In our model the number of viral particles that initiate infection is a random variate drawn from the number of viruses to which a person is exposed. This means that, even given a very tiny exposure, there is some small risk of the person being infected. However, as the exposure increases, and involves more and more viral particles, there is an increased chance that more viruses will initiate infection. As more viruses initiate infection, the genetic diversity of the viruses founding infection will, on average, increase.

Genomic data has shown that for influenza and SARS-CoV-2 infection, the transmission bottleneck generally involves few viral particles^{13–16}, often only one.

We introduce a term called the effective viral load, which describes the number of viruses that would be expected to cause an infection per ml of emitted material inhaled. This term in some way circumvents any discussion of what proportion of viruses cause infection: It is defined as the number that do cause infection.

The effective viral load in our model describes the number of viruses per ml of emitted material that are expected to overcome the various barriers, whether physical or immunological, to initiate infection in the recipient.

We discuss at length the ratio between the number of viruses in emitted material and the effective viral load. The exact ratio is probably very difficult to measure, but experimentally it is possible to measure a) the number of copies of viral genome per ml of sample collected from a host, for example via nasal swab. This may not be the same concentration as would be emitted, but is an approximation; b) The number of plaque-forming units per ml of sample; c) The number of focus-forming units per ml of sample. The difference between b and c is essentially that c) describes the concentration of viruses that have the potential to infect cells, while b) describes the concentration of viruses that have the potential to infect cells so that those cells productively infect other cells. We argue that the ability to form a plaque under favourable experimental conditions is necessary for a virus to initiate infection, and that a ratio of 3000:1 genomes per ml to effective viruses per ml is reasonably conservative. We note the substantial controversy around minimal infectious doses expressed as PFU, citing the review of SeyedAlinaghi et al.

An important source of uncertainty in our model is the relationship between the raw numbers of viral particles contained in emitted material, the number of plaque-forming units (PFU) this represents, and the true effective viral load. The raw numbers of viruses in emitted material, known as the viral load, follows a pattern of growth then decay during the course of infection, which in SARS-CoV-2 infection reach a peak potentially of 10^{10} viruses per ml⁴³. In our default model we have used parameters from one study describing SARS-CoV-2 infection, which suggest a 3,000-fold ratio between the raw number of viruses and the number of PFUs³⁵, further assuming a 1:1 ratio between PFUs and effective viral load. However, we note considerable variation in the literature. Experimental work has suggested a strain-dependent ratio between SARS-CoV-2 viral load and focus-forming units, a measure in some ways similar to PFU, of between 10^4 :1 and 10^6 :1⁴⁴, while a challenge study of SARS-CoV-2 infection estimated close to a 10^5 :1 ratio⁴⁵. The ratio between PFU and the TCID₅₀, the dose needed to initiate infection in 50% of individuals, is a topic of considerable controversy, with a review of the subject identifying estimates spanning several orders of magnitude, from 1.26 to 7×10^6 .²⁵ PFU⁴⁶. Modelling studies have attempted to estimate directly the ration between raw and effective viral loads, with a study of superspreading events concluding on the basis of a Wells-Riley model suggesting a ratio of between 2000:1 and 300:1⁴⁷. If we assume that the ability to form plaques under favourable experimental conditions is a necessary condition for a virus to cause infection in a host, and allow for flexibility given the assumptions underlying modelling studies, our 3000:1 ratio is likely at the conservative end of the spectrum.

Minor Comments:

-In my experience, a more commonly used term in place of viral “degradation” would be viral “inactivation.” This is prevalent in viral exposure modeling research and I think will be more interpretable by those audiences.

Done

-In Table 1, the references look like exponents, and I found the values very surprising at first.

Ha! Perhaps having reference columns per parameter would be useful here for clarity.

We agree that this could be confusing. We have adopted a convention of putting a space in between a numerical value and the following reference e.g. 10^2 is 100 but 10^2 means that we got the value 10 from reference 2.

-Line 13: “found an infection” – like initiate an infection? The word “found” threw me off here.

We agree that “initiate” is a less ambiguous word here.

-Line 59: “Ventilation removes particles from the air” – or dilutes, depending on the ventilation system. I think this should be clarified since ventilation has been such a contentious and poorly understood topic for some during COVID-19.

We agree that there is more than one form of ventilation. We now write:

“ventilation reduces the mean concentration of particles in the air”

-Line 118 – It’s unclear what this parameter means until much further in the Methods at the end of the paper. I recommend a more comprehensive definition at its first mention in the main text.

We have substantially revised the text. This parameter is now only found in Supplementary Information.

-Lines 124-130 read in a confusing way. It’s unclear whether these findings are just due to the model dynamics or if they are anticipated to actually reflect reality or both.

We have substantially altered the way in which we present the results from our diffusion model. After considering different environments under a default emission model, we consider the effect of altering the effective viral load across multiple orders of magnitude, then consider the effects of altering the volume emitted, again across a few orders of magnitude. The first range, across effective viral load, likely reflects a real phenomenon, in so far as the within-host viral load does change across orders of magnitude during infection. The latter range is a little bit more hypothetical, but it is possible to imagine circumstances where a lot of material is emitted by an infected person. Our interest in the latter case is to note the situation in which, if enough material is emitted, a point is reached where exposures are high enough that the majority of people infected receive a large number of effective viruses.

-Line 589, what does “not accessible to our model” mean? In the same line as this, what does “required number of infections could not be obtained for any effective load” in figure 3 mean? Giving more detail about the computational limitation here would be useful for modelers.

What we mean in that case is a situation where the number of people infected is set to be high, in the absence of having a high enough volume of material emitted by the infected person. Not accessible means that we are asking our model to create a situation in which more people were infected than were exposed to a single emitted particle, which is impossible.

This situation is no longer relevant given the way we have rewritten the manuscript.

-Figure 3 – labels for the color heatmap bars would be useful so that not all of the figure title/caption text has to be read to interpret.

This figure no longer exists as we have remade the vast majority of the Figures.

Reviewer #3 (Remarks to the Author):

It is a useful attempt to derive the bottleneck size for a respiratory virus using an alternative approach than the commonly used genome sequencing data. I am glad that the authors tried to do it. However, there appear several major issues with the mathematical models used in the current study and the choice of the studied settings.

I also have some concerns over the conclusion that it is airborne transmission that causes the tight transmission bottleneck, which is probably not fully supported by the data so far in the paper. I will probably focus on the method aspects. Hope that these questions may help the authors to refine their model.

Here we are saying that tight bottlenecks would be expected for most infections as a consequence of airborne transmission. The exceptions to this rule are a) when the effective viral load is so high that the inhalation of a single emitted particle causes infection with multiple viruses and b) when the level of exposure is so high that a high proportion of individuals present are infected with multiple viruses (and therefore an even higher proportion of individuals are infected).

First, the effective viral load k_b is probably the most important and crucial parameter. In this study, its value is calculated to match a specific R value (Line 552-553). This is right. The authors probably assumed that everything in their model was correct, so that the model could back calculate the number concentration of infectious virus particles (virions) in the expired fluid. There is no validation of the back calculated number concentration of infectious viruses in the expired fluid (to my knowledge, such data also does not exist for the general population).

We agree that the effective viral load is very important. In this rewrite of the manuscript we show the majority of our results on a scale of effective viral load spanning very low to very high values.

First of all, we calculate a default level for the effective viral load. There is no one 'correct' value for this parameter, as the raw number of viral genomes per ml changes by orders of magnitude during the course of infection. We use an epidemiological model to infer a default value which might be too high, but is within the realms of feasibility. We next explore different values of this parameter.

As discussed in the response to reviewer 2, we don't think this value has been directly experimentally validated. Ideally, we would want combined measurements of the number of genomes per ml of expired material e.g. collected from coughing, in conjunction with measurements of plaque forming units for the same material, collected from multiple individuals spanning a population. In the absence of these precise data we consider what data are experimentally available to estimate a plausible range for this parameter.

It is also known that expired droplets of varying sizes may contain various number of virions (viruses or particles as in Line 528-554). It appears that the possibility of droplets containing multiple infectious viruses is not dealt with in this paper.

We explicitly allow for droplets to contain multiple infectious viruses. In our simulation each person is exposed to a stochastic number of droplets of various sizes according to the calculated level of physical exposure. Each droplet contains a stochastic number of effective viruses according to the volume of the droplet and the effective viral load.

The authors could have used the existing viral load data in the literature, then the crucial dose-response parameter is needed.

The viral load varies by orders of magnitude across the course of infection. What is relevant here is the effective viral load, which is the product of the number of viral genomes multiplied by the proportion of those genomes which go on to initiate infection. This proportion is itself unknown. Experimental measurements of the proportion of viruses which form plaques or foci of infection differ by orders of magnitude. Also, the number of plaque forming units which are required to initiate infection is unknown, with estimates spanning orders of magnitude. Our approach is to use a value from the literature which we believe can be justified as conservatively high.

I do not fully understand the rationale behind the choice of R_{office} value – an $T_{office} = 0.02675$ was chosen. As $R_0=5$ is chosen, so that R_{office} is around 0.1. I do not understand about the rationale of this approach – do you mean that you have considered the situation in a “typical” office that you considered, R_{office} (as used in equation Line 550) equals to 0.02675×0.5 ? This does not seem right to me.

We use an epidemiological model to set the default effective viral load. The value R_{env} describes an expected number of infections in the environment in question. This implies a value for the effective viral load. This is now better explained in Supplementary information.

By the way, for the ancestral strain of SARS-CoV-2, I recall that most cited R_0 is about 2-2.5 (a systematic review of literature is needed), not 5 (Line 588)

We now use a value of 2.5.

Second, I have major concerns about the use of a 1D diffusion model (Line 308). There is no validation of the predicted spatial expired particle (droplets) with any experimental data. Only the final bottleneck size data are presented. It is almost impossible to judge or reproduce the model.

We have completely rewritten the model, using a 2D model of diffusion with the removal of emitted particles. The third dimension of room height is accounted for in so far as we model sedimentation and represent height in an estimate of particle concentration by volume. The model is based upon published parameters from experimental measurements of emissions e.g. from coughing and speaking. We explore different values of these parameters in different environments and also conduct sensitivity analyses on these parameters.

The model is reproducible in so far as we make our code publicly available with instructions for calculating bottleneck sizes. We are not sure how to further validate our model experimentally: We could hypothesise building a room, then measuring the concentrations of particles resulting from someone coughing, but our model describes an idealised environment, and does not attempt to capture specific factors such as ventilation that a more complex, but less generalisable approach, such as computational fluid dynamics, would do.

The 1D means that author needs to choose a dimension x – (width or length of the room, presumably in front of the index case, what do you do with the space behind the index case), and then consider a radius of exposure x (Line 490) for a susceptible individual at a location (Line 474). The authors were also able to consider the boundary conditions at walls (again I am not sure how the four walls can be considered in a 1D simulation, except if you do the calculation in all directions (360 degree around the infected, I also guessed that the authors did this).

This correct, and is a key reason why we adapted our model to describe a 2D environment.

This might be a very crude assumption as the turbulent diffusion approach is not good at modelling the expired puffed jets.

A 2D model better captures the behaviour of emitted particles following a cough. We model a separation of timescales: The expiration via coughing or sneezing is modelled as instantaneous, while the subsequent turbulent diffusion occurs over the subsequent hour.

Third, many model decisions and parameters used are not valid to me. The assumption that the infectious virus particles are only released by cough is problematic. It is now known that breathing also emission particles. I cannot imagine how a cough (10 times per unit time) without mouth covering be possible in some of the modelled environment?

Previously we modelled what would happen in the case of particles being emitted via coughing. We now also model speaking and sneezing. The key parameters in our emission model are the distribution of emitted particle sizes, the initial location of particles following emission, and the total volume emitted. The distribution of particle sizes was taken from published models, while a large range of volumes is explored in a sensitivity analysis.

The cough droplets can also be bimodal, not a single mode lognormal (Line 319).

The model we took from the literature was unimodal, but sneezing was modelled as having a bimodal distribution.

I also do not know why an artificial 80 micron was assumed as the largest droplets, as larger ones were found. Patients do not sneeze – not sure why sneeze is also considered (Line 346).

We now model particles up to 500 micron radius for coughing and speaking and up to 1000 micron radius for sneezing.

Sneezing was considered in order to evaluate the sensitivity of our results to a different emission distribution: We potentially have in mind emissions from respiratory infections other than SARS-CoV-2. Although the sneeze model is radically different to that of coughing and speaking (we base this on parameters taken from different parts of the literature), the basic result is the same: Bottleneck sizes are small with the two exceptions noted for coughing.

What was the rationale of choosing final size after evaporation as $r_0/4$ (Line 369)?

This is an approximation. We reference Stiti et al., who show for example a droplet of $20\mu\text{m}$ evaporating to $4.7\mu\text{m}$, and a droplet of $257\mu\text{m}$ evaporating to $60\mu\text{m}$.

Would the simple diffusion measurement in ref 50 be applicable to all four venues considered in this study?

The measurement provides us with an approximation for the rate of turbulence diffusion given the size of the environment and the number of room changes per hour. We are aware that it is an approximation, and so have carried out a sensitivity analysis of the effect of changing the rate of turbulence diffusion in our model; this is shown in Supplementary Figure S7.

I am suspicious of the typical air change rates (Table 1) including using the data from Engineering Toolbox (some data without reference there). One big question to ask – if these are typical air change rates, would those outbreaks also occurred in settings with these air

change rates, if not would your estimate of kb value with R0-5 be suspicious?

The inhalation/exhalation rate is 10 in office, and 12.5 on bus – I would have thought that passengers would sit quietly and in an office environment, workers have a greater met value.

As noted, our rates of turbulence diffusion are approximations, and we carry out a sensitivity analysis on this parameter. We are not completely certain what is or is not suspicious, but the number of people infected is complex, depending upon the extent of crowding in a room and the manner of the emissions from the infected person alongside the ventilation rate. Our concern here is the distribution of transmission bottleneck sizes, which varies in a manner that we describe in this manuscript.

Regarding the inhalation/exhalation rate, the value for the bus was taken directly from the literature. How people behave in an office could vary from place to place. As discussed in our response to reviewer 1, varying the inhalation rate is equivalent to varying the volume of particles emitted, altering the exposure by a linear factor. We now explicitly consider the effect of changes in the volume of emissions upon the distribution of transmission bottleneck sizes.

I stop here and I may suggest authors to consider a few real outbreak settings where detailed ventilation data is available.

We thank the reviewer for their comments. Our understanding is that a computational fluid dynamics model would capture the details of ventilation in any given environment, and that such a model would show some distortion to the more even distribution of exposure the results from our approach. Distortion of this form would probably make the extent to which people were exposed more highly variable. Our model of a bus captures a situation in which the exposure distribution varies by several orders of magnitude. In this sense, we have implicitly explored the sensitivity of our model to variation in the exposure distribution: The results from the bus are more complex than the other environments, with for example no situation arising in which everyone on the bus is infected, but the pattern of transmission bottleneck sizes is broadly replicated: There is no major change here that makes us believe that the lack of detailed ventilation modelling substantially alters our main result.

A very recent publication has described the risk of transmission in digitally-detected cases of contacts, with potential implications for our default parameter settings: It is likely that our default effective viral load is too high. Although given our consideration of a broad range of potential effective viral loads this does not affect our results, we have added a note to the discussion, with further detail in section 2.2 of supplementary text.

Our model is not intended to capture the full complexities of viral transmission that would facilitate direct comparison with real-world scenarios. A recent publication suggests an infection risk of slightly under 10% for individuals in the most risky scenarios after 8 hours of exposure⁴³, which compares to 18.7% or 74.9% in our default lounge models of coughing and continuous speech: As discussed in Supplementary Text, our default parameters likely overestimate the effective viral load in a realistic situation. We note that a complete accounting for transmission would require an account of the precise distributions of emissions, viral load, and time-dependent proximity, alongside environmental parameters and human behaviour.

REVIEWERS' COMMENTS

Reviewer #1 (Remarks to the Author):

I thank the authors for the attention to mine and other reviewers comments.

I think the authors are dancing on a pinhead a little about QMRA in that (in my understanding) the point of QMRA is to take an exposed dose and infer the chance of infection. The authors make an case in their response that is compelling and I think this is a bit of a cul-de-sac given the focus of paper so I will leave this here.

I am happy to accept the proposed changes.

My only lingering query on rereading relates to Eqn 1. The Haas QMRA text book and associated literature offers extension away from the exponential dose response (eqn 1) allowing say variation in clearance/innate immune response between individuals or over-dispersion in dose which turns the exponential relationship to a Beta-Poisson form (not to be confused with the approximate-beta Poisson that is the focus of the aforementioned text book.) or more complicated hypergeometric functions. Do the authors think the Poisson assumption is particularly sensitive? A little test of overdispersion via beta- or gamma-Poisson structures may be interesting.

Reviewer #2 (Remarks to the Author):

Thank you for a thorough response to my comments on the last version. I think this version is much improved but is still difficult to read. For example, the Results right now read as a series of explorations, where one exploration leads to another. It is overall hard to follow, and I think a visual showing the overall approach/explorations that readers can use to follow along would be extremely helpful. Perhaps a table that describes the research question and model assumptions for each endeavor would work for this purpose. The use of subsections would also help in the Results and Discussion.

Reviewer #4 (Remarks to the Author):

Dear authors,

Thank you for addressing Reviewer #3's comments, you have made substantial effort to I feel you have made a good call on remodelling the environments using a 2D model.

We thank the reviewers for the comments, and respond below.

Reviewer #1 (Remarks to the Author):

I thank the authors for the attention to mine and other reviewers comments.

I think the authors are dancing on a pinhead a little about QMRA in that (in my understanding) the point of QMRA is to take an exposed dose and infer the chance of infection. The authors make an case in their response that is compelling and I think this is a bit of a cul-de-sac given the focus of paper so I will leave this here.

I am happy to accept the proposed changes.

My only lingering query on rereading relates to Eqn 1. The Haas QMRA text book and associated literature offers extension away from the exponential dose response (eqn 1) allowing say variation in clearance/innate immune response between individuals or over-dispersion in dose which turns the exponential relationship to a Beta-Poisson form (not to be confused with the approximate-beta Poisson that is the focus of the aforementioned text book.) or more complicated hypergeometric functions. Do the authors think the Poisson assumption is particularly sensitive? A little test of overdispersion via beta- or gamma-Poisson structures may be interesting.

We agree that overdispersion is an important factor relative to a Poisson model: In any realistic scenario there will be different levels of exposure. Our physical model of particle dispersion aims to capture the kinds of differences in exposure level that would be expected in a realistic environment, with the caveats noted in the Discussion. We note that our physical model captures behaviour, such as the non-independence of virus particles due to their emission within droplets of non-negligible size, that are not captured even by an extension to a beta-Poisson or gamma-Poisson model.

Nevertheless, we have repeated the analysis using Equation 1 to include a gamma-Poisson model. While there are differences in the numerical values of some results the overall picture is not greatly changed. We have added the new Supplementary Figure 1 showing the results, which are comparable to those of Figure 1.

Reviewer #2 (Remarks to the Author):

Thank you for a thorough response to my comments on the last version. I think this version is much improved but is still difficult to read. For example, the Results right now read as a series of explorations, where one exploration leads to another. It is overall hard to follow, and I think a visual showing the overall approach/explorations that readers can use to follow along would be extremely helpful. Perhaps a table that describes the research question and model assumptions for each endeavor would work for this purpose. The use of subsections would also help in the Results and Discussion.

We have added headings to demarcate the Results section. We agree that a lot of our results represent explorations of the sensitivity of our results to different parameters, largely performed in response to queries from reviewers. Moving these to Supplementary Text has both shortened and simplified the Results section. With this simplification we think the structure of the manuscript is now more accessible, so as to not require an additional Table or Figure.

Subsections are not allowed in the Discussion under journal policy; we have made minor edits to improve readability.

Reviewer #4 (Remarks to the Author):

Dear authors,

Thank you for addressing Reviewer #3's comments, you have made substantial effort to I feel you have made a good call on remodelling the environments using a 2D model.

We thank reviewer 4 for the comments received.